# Propensity-driven Uncertainty Learning for Sample Exploration in Source-Free Active Domain Adaptation

## Abstract

Source-free active domain adaptation (SFADA) addresses the challenge of adapting a pre-trained model to new domains without access to source data while minimizing the need for target domain annotations. This scenario is particularly relevant in real-world applications where data privacy, storage limitations, or labeling costs are significant concerns. Key challenges in SFADA include selecting the most informative samples from the target domain for labeling, effectively leveraging both labeled and unlabeled target data, and adapting the model without relying on source domain information. Additionally, existing methods often struggle with noisy or outlier samples and may require impractical progressive labeling during training. To effectively select more informative samples without frequently requesting human annotations, we propose the Propensity-driven Uncertainty Learning (ProULearn) framework. ProULearn utilizes a novel homogeneity propensity estimation mechanism combined with correlation index calculation to evaluate feature-level relationships. This approach enables the identification of representative and challenging samples while avoiding noisy outliers. Additionally, we develop a central correlation loss to refine pseudo-labels and create compact class distributions during adaptation. In this way, ProULearn effectively bridges the domain gap and maximizes adaptation performance. The principles of informative sample selection underlying ProULearn have broad implications beyond SFADA, offering benefits across various deep learning tasks where identifying key data points or features is crucial. Extensive experiments on four benchmark datasets demonstrate that ProULearn outperforms state-of-the-art methods in domain adaptation scenarios.

## 1 Introduction

In recent years, convolution neural networks have achieved remarkable success across a wide range of computer vision tasks, including image classification, object detection, and semantic segmentation. However, the performance of these models often degrades significantly when faced with substantial domain shifts between training and testing distributions. This degradation is particularly problematic in real-world applications where we need to transfer a pre-trained model from a source domain to different downstream target domains. Moreover, obtaining human annotations for large datasets or new domains is time-consuming and expensive. To address these challenges, researchers have proposed various unsupervised domain adaptation (UDA) techniques (Long et al., 2018; Xu et al., 2019). These methods aim to adapt models trained on a labeled source domain to unlabeled target domains, thereby reducing the need for extensive manual annotation in the target domains.

However, traditional UDA approaches require simultaneous access to both source and target domain data during the adaptation process. This requirement induces several practical challenges, including increased storage and training resource demands as well as privacy concerns associated with sharing or storing source domain data. To overcome these limitations, we focus on a more practical setting known as source-free active domain adaptation (SFADA). In this scenario, the model is initially trained on the source domain, but during adaptation, only the target domain data is available. Additionally, the active learning component allows for selective annotation of a small subset of target domain samples to maximize adaptation performance while minimizing labeling costs.

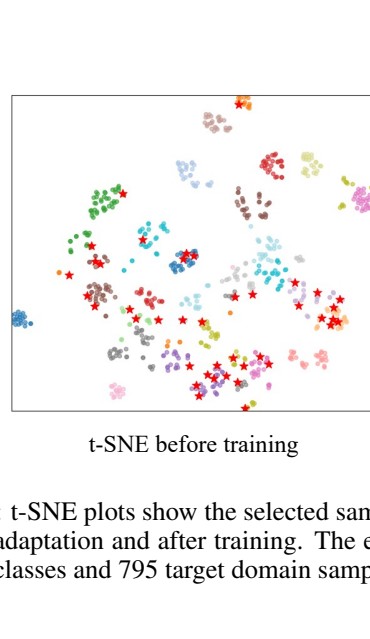 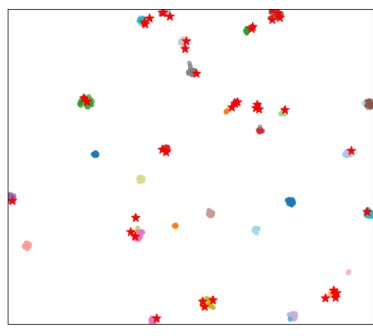

t-SNE before training          t-SNE after training

Figure 1: t-SNE plots show the selected sample location (red stars) and overall feature space before domain adaptation and after training. The experiments are based on the Office-31 dataset (A→W) with 31 classes and 795 target domain samples.

The active learning aspect of SFADA is particularly significant, as it addresses a fundamental challenge in machine learning: identifying the most informative data points or features for model training and adaptation. This challenge extends far beyond domain adaptation, influencing areas such as medical image analysis, natural language processing, and robotics. By developing more sophisticated methods for informative sample selection, we can improve model efficiency, reduce data requirements, and enhance performance across a wide range of deep learning applications.

Current methods in this field (Li et al., 2022; 2023b; Du & Li, 2023) normally utilize techniques such as pseudo-labeling, self-training, or adversarial learning to leverage the knowledge encoded in the pre-trained source model and adapt it to the target domain with minimal supervision. They often select active samples where the model has high predictive uncertainties. However, these samples have a great probability of belonging to noise or outliers of the class, which can influence the prediction of neighbor samples and lead to performance degradation during adaptation. In addition, some of the active domain adaptation methods (Xie et al., 2022; Sun et al., 2023; Du & Li, 2023) progressively request human annotation during the training process, which is not practical and inconvenient.

To improve the domain adaptation efficacy and address existing challenges on informative sample selection without frequently requesting human annotations, we propose a Propensity-driven Uncertainty Learning framework (ProULearn). This framework incorporates informative sample selection, pseudo-labeling, and correlation alignment for domain adaptation. At its core, ProULearn utilizes a homogeneity propensity estimation mechanism and correlation index to evaluate the grouping condition and model prediction confidence of each data point. This approach aims to select samples that are grouped with others with low prediction confidence, enabling the model to learn the target domain information it currently lacks. Our framework further enhances performance by combining the homogeneity propensity score with correlation relationships of samples to different class centroids, facilitating accurate pseudo-label assignment. To improve class distinction, we introduce a central correlation loss that encourages the model to concentrate on class centroids, resulting in more compact and easily distinguishable class clusters. The effectiveness of our approach is visually demonstrated in Figure 1, which presents t-SNE visualizations (Van der Maaten & Hinton, 2008) of the data distribution before and after adaptation. Initially, most of the selected active samples (red stars) are shown to be strategically grouped with other samples in high data density areas. Following the adaptation process, we observe a significant concentration of most clusters around these selected samples, illustrating the model's successful learning from these key data points and its improved domain adaptation capabilities. The contributions of our work can be summarized as follows:

- We propose a novel ProULearn (Propensity-driven Uncertainty Learning) framework that addresses the challenges of source-free active domain adaptation tasks. ProULearn integrates three key components: informative sample selection, pseudo-labeling, and correlation alignment, to effectively learn target domain features in cross-domain scenarios.

- An innovative sample selection mechanism is developed within ProULearn, which combines homogeneity propensity estimation and correlation index calculation. This mecha-

nism evaluates both the grouping condition of data points and the model's prediction confidence. By selecting samples that are grouped with others having low prediction confidence, our method enables the model to focus on and learn from the most informative examples in the target domain, thereby enhancing its ability to adapt to new domains.

- The proposed ProULearn framework successfully addresses the challenges of SFADA, demonstrating state-of-the-art performance across multiple benchmark datasets. Moreover, the principles underlying our approach to informative sample selection offer insights that could benefit active learning strategies in various deep learning tasks.

## 2 RELATED WORKS ON ACTIVE LEARNING

Active learning is getting great attention nowadays. Compared with traditional supervised learning strategies which heavily rely on human annotations, active learning aims to utilize the pre-trained model on unlabeled downstream tasks or domains with selective training samples. It is a more practical training scene which greatly reduces the annotation costs and time. People adopt active learning concepts in various deep learning tasks including image or text classification (Kim et al., 2021; Seo et al., 2022), object detection (Su et al., 2020; Yuan et al., 2021), person re-identification (Liu et al., 2019; Teng et al., 2023), etc. Recently, people starting to introduce active learning into domain adaptation tasks to reduce the labeling cost for transferring the model from one domain to another (Prabhu et al., 2021; Mathelin et al., 2022; Xie et al., 2023). Sun et al. (2023) progressively requested and augmented active samples as well as their expanded neighbour to refine the network. Xie et al. (2022) alleviated the domain gap by using regularization terms. Meanwhile, it incorporates domain characteristics and instance uncertainty into the active sample selection process.

Recently, source-free domain adaptation (SFDA) has gathered some attention. It aims to transfer a pre-trained model from a source domain to target domains without access to the source domain data. A more detailed introduction to SFDA can be found in the Appendix. While this approach offers potential benefits in reducing annotation efforts during adaptation, current research has encountered challenges in achieving substantial improvements. Given these constraints, our paper focuses on a more practical and effective paradigm: source-free active domain adaptation (SFADA). This strategy incorporates active learning techniques to enhance model performance and adaptability.

## 3 METHOD

### 3.1 PROBLEM FORMULATION

Source-free active domain adaptation (SFADA) addresses the challenges of adapting a pre-trained model from a source domain $\mathcal{S}$ to a target domain $\mathcal{T}$ without access to the original source data. In the source domain, we have a labeled dataset $\mathcal{S} = (\mathcal{X}_s, \mathcal{Y}_s)$ used to train the initial model under supervision. The adaptation phase focuses solely on the target domain $\mathcal{T}$, which consists of a small set of labeled data $\mathcal{T}_l$ and a large set of unlabeled data $\mathcal{T}_u$. The complete target domain is thus $\mathcal{T} = \mathcal{T}_l \cup \mathcal{T}_u$. The key constraint in SFADA is the limited labeling budget for $\mathcal{T}_l$, which is expressed as a percentage $B\%$, determining the size of $\mathcal{T}_l$. This setting presents two major challenges: (1) selecting the most informative samples from $\mathcal{T}_u$ for labeling as $\mathcal{T}_l$ to maximize adaptation effectiveness; (2) leveraging both labeled and unlabeled target data to adapt the model effectively.

To effectively select informative samples and adapt the pre-trained model to the target domain, we proposed a Propensity-driven Uncertainty Learning (ProULearn) framework to address the SFADA tasks. The overall training process can be found in Figure 2. In the following sections, details of selecting samples, pseudo-labeling, and training process are introduced.

### 3.2 INFORMATIVE SAMPLE SELECTION

Our goal is to select samples that are grouped together and the model has low confidence in identifying them. These samples contain the most signature features from the class from which the model can learn the class's representative features. Since the model has low confidence in identifying them, there exists domain gaps between the model's knowledge with the class's features. By annotating

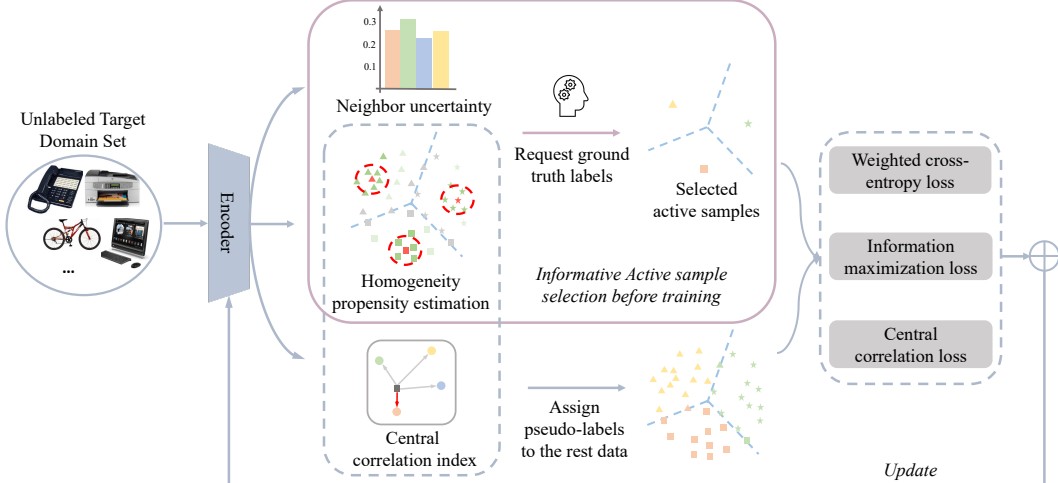

Figure 2: Overall ProULearn framework during target domain adaptation. The informative samples are selected before the training using HPE and correlation entropy. These active samples are fixed during the adaptation process. Meanwhile, the model and pseudo labels are refined during training.

these samples, the model can focus on the most important features of the class, thus better adapting to the target domain and providing more accurate pseudo-labels for other non-labeled samples.

**Homogeneity propensity estimation.** To effectively identify the grouping condition of each sample from a global perspective, we introduce a homogeneity propensity estimation mechanism (HPE) inspired by the isolation forest algorithm (Liu et al., 2008). The algorithm is first introduced for abnormal data detection and we develop our HPE inspired by its structure to assign a homogeneity score to each sample. Our approach offers advantages over traditional unsupervised clustering methods, such as K-means (Lloyd, 1982) or HDBSCAN (Campello et al., 2013), which rely heavily on initial distributions and local cluster relationships. The HPE mechanism can capture the sample grouping condition from the whole training set's perspective and is particularly valuable when dealing with large domain gaps, where initial distributions may be inaccurate.

The HPE mechanism begins by constructing an ensemble of $g$ separation trees. Each tree is built using a random subset of the data. At each node of the tree, a random feature $m$ is selected, and a split value $v$ is randomly chosen between the minimum and maximum values of the selected feature in the current subset. The data is then split into left and right child nodes based on whether their value for feature $m$ is less than or greater than $v$. As shown in Figure 3, this process continues until a maximum depth is reached or a node contains only one sample. Each node stores the depth in the tree, the feature index used for splitting, and the split value. The splitting process ensures that the tree adapts to the local density of the data, as the split value is chosen uniformly between the minimum and maximum values of the selected feature in the current subset. This allows the tree to create more splits in regions of high density and fewer splits in regions of low density. Consequently, samples in dense regions typically have longer path lengths, reflecting their homogeneity, while samples in sparse regions have shorter path lengths, indicating their potential anomalous nature.

The maximum depth of the tree is set to $r_{max} = \log_2(\text{sample\_size})$. This depth limit prevents overfitting by avoiding unnecessarily deep trees that might capture noise rather than true data patterns while allowing the tree to capture the essential structure of the data. The logarithmic relationship ensures that the tree depth scales reasonably with the sample size without increasing model complexity. This design effectively handles datasets of varying sizes without manual tuning.

The homogeneity score $h(x)$ for each sample is calculated as the average path length across all trees:

$$h(x) = \frac{1}{g} \sum_{i=1}^{g} r_i(x), \tag{1}$$

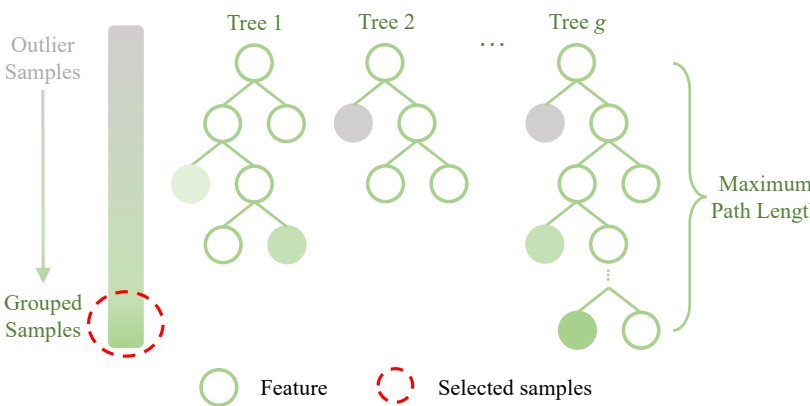

Figure 3: Principle of the homogeneity propensity estimation mechanism. An ensemble of trees is used to estimate sample homogeneity, where longer paths indicate more grouped samples. The mechanism selects representative grouped samples (green circles) rather than outliers (grey circles), guiding the model to focus on these samples during adaptation. This approach results in more compact and separable class distributions, as evidenced by the results shown in Figure 1.

where $r_i(x)$ is the path length of sample $x$ in the $i$-th tree, and $g$ is the total number of trees created. This averaging process helps to stabilize the score and reduce the impact of any single tree's potentially biased structure. In this approach, samples with shorter average path lengths are considered less homogeneous (more anomalous), while samples with longer average path lengths are considered more homogeneous (more normal). This is because homogeneous samples are expected to require more splits (longer paths) to be isolated. The intuition behind this is that anomalous points are few and different, and thus should be easier to separate from the rest of the samples.

The proposed HPE mechanism provides a robust way to assess the grouping condition of samples, taking into account the global structure of the data while being less sensitive to initial distributions. It doesn't rely on distance calculations or density estimates, which makes it particularly valuable for SFADA tasks, where the initial distribution may be inaccurate due to large domain gaps.

**Correlation index calculation.** To further refine our sample selection process, we introduce a correlation index for evaluating two samples' distribution. For assigning pseudo-labels, we utilize the index to help capture the relationship between samples and their neighbor representations in the feature space. The process begins by computing the correlation matrix between all pairs of samples. For each sample $x_i$, we calculate its correlation with every other sample $x_j$ via:

$$C(f(x_i), f(x_j)) = \frac{\sum_{d=1}^{D}(f(x_i)^d - \bar{f}(x_i))(f(x_j)^d - \bar{f}(x_j))}{\sqrt{\sum_{d=1}^{D}(f(x_i)^d - \bar{f}(x_i))^2}\sqrt{\sum_{d=1}^{D}(f(x_j)^d - \bar{f}(x_j))^2}}, \quad (2)$$

where $D$ is the dimension of sample $x_i$'s feature embedding $f(x_i)$, and $f(x_i)^d$ is the $d$-th feature of sample $x_i$, and $\bar{f}(x_i)$ is the mean of all features for sample $x_i$. This correlation index $C$ provides a measure of distribution between samples based on their feature representations. After computing the correlation indices, we select the $K$ nearest neighbors with the largest correlation values for each sample. This means we select samples that are most similar or positively correlated to the given sample. By selecting neighbors based on high correlation values, we focus on samples that share similar patterns or characteristics in the feature space. For these $K$ nearest neighbors, we then calculate the average output probability and its entropy:

$$E_i = -\sum_{c=1}^{C}\bar{p}_c(x_i)\log(\bar{p}_c(x_i) + \epsilon), \quad \bar{p}(x_i) = \frac{1}{K}\sum_{k=1}^{K}p(x_i)_k, \quad (3)$$

where $\bar{p}_c(x_i)$ is the average probability of class $c$ for the $K$ nearest neighbors of sample $x_i$. $p(x_i)_k$ is the output probability for the $k$-th nearest neighbor of sample $x_i$ and $\epsilon$ is a small constant to prevent

logarithm of zero. By calculating the entropy of the average predictions of a sample's neighbors, we gain insight into the model's certainty about that sample's class. Low entropy indicates consistent predictions among neighbors, suggesting higher confidence, while high entropy suggests disagreement or uncertainty. In SFADA, where we lack source domain data, leveraging local neighborhood information helps ensure that predictions are consistent within similar regions of the feature space. By identifying regions of high and low entropy in the target domain, we can focus adaptation efforts on areas where the model is uncertain.

The final sample selection score $U_i$ combines the homogeneity score and the correlation index:

$$U_i = h(x_i) \times E_i, \tag{4}$$

where $h(x_i)$ and $E_i$ are the normalized homogeneity score and entropy respectively. We select samples with larger $U_i$ scores through an iterative process that ensures diversity. After selecting a sample with the highest $U_i$ score, we exclude its K-nearest neighbors from subsequent selection considerations to avoid redundant selections. This neighborhood-aware selection strategy ensures that selected samples are not only representative of their local regions (thus likely to be representative of a broader group of samples) but also well-distributed across different regions of the feature space. The homogeneity ensures that the selected sample is not an outlier but rather a good representative of a cluster of data points in the target domain. Meanwhile, the high entropy of predictions for its neighbors suggests that this is an area where the model currently lacks discriminative power.

By focusing on these samples, we aim to provide the model with the most informative examples from the target domain. This streamlined approach to sample selection allows the model to improve its performance on the target domain within the budget of labeled examples. Also, the selection process is done before the training process, which is ideal for practical applications since the model will not progressively acquire human annotation during the training process.

### 3.3 PSEUDO-LABELING AND REFINEMENT

After selecting the most informative samples, we assign pseudo-labels to the remaining unlabeled samples. We begin by initializing class centroids in the feature space as in (Liang et al., 2020; Wang et al., 2023). The class centroids $o_c$ are derived from the model's current class predictions:

$$o_c^j = \frac{\sum_{x_i \in \mathcal{X}_t} \delta_j(p(x_i)) \cdot f(x_i)}{\sum_{x_i \in \mathcal{X}_t} \delta_j(p(x_i))}, \tag{5}$$

where $\delta_j(p(x_i))$ is the model's $j^{th}$ logit output $p(x_i)$ of sample $x_i$ belonging to class $c$, and $f(x_i)$ is the feature vector of sample $x_i$. To obtain and refine the pseudo-labels, we incorporate the homogeneity scores into the decision process. The correlation index between each sample's feature vector and the class centroids is calculated via Eq. 2, then multiply this correlation index with the homogeneity scores by:

$$z_{i,c} = C(f(x_i), o_c) \cdot h(x_i), \quad \forall c \in \{1, ..., M\},$$
$$\hat{y}_{tu} = \arg\max_c z_{i,c}, \tag{6}$$

where $z_{i,c}$ is the similarity score between sample $x_i$ and class centroid $o_c$, $C(\cdot)$ denotes the correlation index, $h(x_i)$ is the homogeneity score for sample $x_i$, and $M$ is the total number of classes. The correlation index is computed for each class centroid $o_c$. The pseudo-label $\hat{y}_{tu}$ for each unselected sample is then assigned based on the highest refined similarity score across all classes.

This approach leverages both the model's current predictions and the structural information captured by the homogeneity scores. Samples with higher homogeneity scores are given more confidence in the pseudo-labeling process, as they are more likely to be representative of their local neighborhoods. Conversely, samples with lower homogeneity scores, which can be outliers, have less influence on the pseudo-labeling decision. By incorporating the homogeneity scores, we aim to produce more reliable pseudo-labels that align with the underlying structure of the target domain data.

### 3.4 TRAINING PROCEDURES

For the source domain supervised learning, we utilize the same method as in (Liang et al., 2020; Yang et al., 2021; Pan et al., 2024b), which adopted cross-entropy loss with label-smoothing to pre-train the model. During the target domain adaptation, the pseudo-labels are refined during the adaptation process using the same strategy as described in Section 3.3, which adopts the central correlation index and homogeneity score to assign pseudo-labels to unlabeled samples. The selected sample label remains unchanged. In our adaptation process, we employ the same weighted cross-entropy loss and information maximization losses to account for the varying confidence in our labels. These loss functions are widely used in domain adaptation tasks (Liang et al., 2020; Wang et al., 2023; Lyu et al., 2024) which are designed to differentiate between actively selected samples and those assigned pseudo-labels. The weighted cross-entropy loss $\mathcal{L}_{wce}$ and information maximization $\mathcal{L}_{im}$ for a batch of samples is calculated as follows:

$$\mathcal{L}_{wce} = -\mathbb{E}_{x_t \in \mathcal{X}_t} \sum_{c=1}^{M} w(x_t) \cdot y_t^c \log(p_c(x_t)), \tag{7}$$

$$\mathcal{L}_{im}(\mathcal{X}_t) = \sum_{c=1}^{M} \mu_c \log \mu_c - \mathbb{E}_{x_t \in \mathcal{X}_t} \sum_{c=1}^{M} p_c(x_t) \log p_c(x_t), \tag{8}$$

where $M$ is the number of classes, $w(x_t)$ is the weight assigned to each sample, $y_t^c$ is the one-hot label for the sample, $p_c(\cdot)$ is the output logit, and $\mu_c = \mathbb{E}_{x_t \in \mathcal{X}_t} p(x_t)$ is the mean output of the target domain. The weights $w(x_t)$ are determined based on whether the sample was part of the actively selected informative set or was assigned a pseudo-label. For actively selected samples, the weights incorporate their combined scores $U_i$. For pseudo-labeled samples, the weights are determined by the magnitude of their similarity score $z_{i,c}$ to their assigned class centroid. This weighting scheme ensures that highly confident pseudo-label assignments have more influence during training. This $\mathcal{L}_{wce}$ allows us to place more emphasis on the actively selected samples while still allowing the model to learn from the pseudo-labeled samples but with a lower level of influence on the overall loss. The $\mathcal{L}_{im}$ helps maintain prediction balance and exploits the structure of unlabeled target data by encouraging the formation of distinct clusters in the feature space.

To further refine our model's adaptation to the target domain, we propose a central correlation loss $\mathcal{L}_{cc}$. This loss encourages the alignment between sample features and their corresponding class centroids. Compensate with the $\mathcal{L}_{im}$ which utilizes output logits to compute entropy, the $\mathcal{L}_{cc}$ is built on feature level which begins by computing the correlation between each sample's feature vector and all class centroids. We utilize the correlation index formula as in Eq. 2 to calculate the correlation between samples and their corresponding class centroid according to ground-truth active labels or pseudo-label. The correlation values are constrained to the range [-1, 1] and convert these correlation values to distances by subtracting them from 1. The final central correlation loss is then computed as the mean of these relevant distances across all samples in the batch:

$$\mathcal{L}_{cc} = \frac{1}{N} \sum_{i=1}^{N} (1 - C(x_t^i, o_c^i)), \tag{9}$$

where $o_c^i$ is the predicted class centroid for sample $x_t^i$ and $N$ is the number of samples in the batch. The total loss is computed as follows:

$$\mathcal{L} = \mathcal{L}_{wce} + \mathcal{L}_{im} + \mathcal{L}_{cc}. \tag{10}$$

By incorporating this central correlation loss, we encourage each sample's feature representation to align more closely with its predicted class centroid. As shown in the class distribution plot in Figure 1, during the adaptation, the outlier samples of each class will be concentrated to the centroid, from where most active samples are selected. This promotes the formation of more compact and separable class clusters in the feature space, leading to improved classification performance and more effective domain adaptation. The loss penalizes cases where a sample's features are dissimilar to its predicted class centroid, thus guiding the model to learn more discriminative features that

better represent the target domain's class structure. The overall adaptation algorithm and important notations are presented in the Appendix.

# 4 EXPERIMENTS

## 4.1 DATASETS DETAILS

The implementation details are presented in the Appendix. Our experiments utilize four SFADA benchmark datasets as below.

*Office-31* (Saenko et al., 2010) comprises 4,110 images across 31 categories in three domains: **A**mazon, **D**SLR, and **W**ebcam. *Office-Home* (Venkateswara et al., 2017) expands the challenge with 15,588 images across four domains: **A**rtistic, **C**lipart, **P**roduct, and **R**eal-World, each containing 65 classes. *DomainNet-126* (Peng et al., 2019) further elevates complexity with four domains, **C**lipart, **P**ainting, **R**eal, and **S**ketch, encompassing 126 classes each. *VisDA-2017* (Peng et al., 2018) tackles **S**ynthetic-to-**R**eal adaptation across 12 object categories, utilizing synthetic renderings for training and real images for validation and testing.

## 4.2 EXPERIMENTAL RESULTS

To validate the effectiveness of the proposed method, we conduct comprehensive experiments on four domain adaptation datasets that cover a wide range of objects and tasks. The performance of our method is compared with state-of-the-art benchmarks including methods for SFDA, active domain adaptation (active DA), and SFADA tasks, e.g., SHOT (Liang et al., 2020), NRC (Yang et al., 2021), AaD (Yang et al., 2022), PFC (Pan et al., 2024b), SF(DA)$^2$ (Hwang et al., 2024), AADA (Su et al., 2020), TQS (Fu et al., 2021), CLUE (Prabhu et al., 2021), EADA (Xie et al., 2022), DUC (Xie et al., 2023), ELPT (Li et al., 2022), DAPM-TT (Du & Li, 2023), LFTL (Lyu et al., 2024), and MHPL (Wang et al., 2023). The results are presented in Table 1 to Table 3, some of them are obtained from (Zhang et al., 2023). Detailed results of each category for the VisDA-2017 dataset can be found in the Appendix. "SF" in tables denotes source data free, *i.e.,* adaptation without source data. We report top-1 accuracy for each domain shift and overall average performance (Avg). The best average accuracy is marked in bold. "†" indicates that results are obtained using the original code from the published papers. The discrepancy between the reproduced results and the results reported in the published paper may be due to hardware environment differences.

Table 1: Performance (%) on the Office-Home dataset with 5% labeled target domain samples.

| Categories | Method | SF | Ar→Cl | Ar→Pr | Ar→Re | Cl→Ar | Cl→Pr | Cl→Re | Pr→Ar | Pr→Cl | Pr→Re | Re→Ar | Re→Cl | Re→Pr | Avg |
|---|---|---|---|---|---|---|---|---|---|---|---|---|---|---|---|
| SFDA | SHOT | ✓ | 57.1 | 78.1 | 81.5 | 68.0 | 78.2 | 78.1 | 67.4 | 54.9 | 82.2 | 73.3 | 58.8 | 84.3 | 71.8 |
| | NRC | ✓ | 57.7 | 80.3 | 82.0 | 68.1 | 79.8 | 78.6 | 65.3 | 56.4 | 83.0 | 71.0 | 58.6 | 85.6 | 72.2 |
| | AaD | ✓ | 59.3 | 79.3 | 82.1 | 68.9 | 79.8 | 79.5 | 67.2 | 57.4 | 83.1 | 72.1 | 58.5 | 85.4 | 72.7 |
| | PFC | ✓ | 60.0 | 80.9 | 82.7 | 68.8 | 80.0 | 79.5 | 68.8 | 58.5 | 83.0 | 72.9 | 60.9 | 86.1 | 73.5 |
| Active DA | AADA | ✗ | 56.6 | 78.1 | 79.0 | 58.5 | 72.7 | 71.0 | 60.1 | 53.1 | 77.0 | 70.6 | 57.0 | 84.5 | 68.3 |
| | TQS | ✗ | 58.6 | 81.1 | 81.5 | 61.1 | 76.1 | 73.3 | 61.2 | 54.7 | 79.7 | 73.4 | 58.9 | 86.3 | 72.5 |
| | CLUE | ✗ | 58.0 | 79.3 | 80.9 | 68.8 | 77.5 | 76.7 | 66.3 | 57.9 | 81.4 | 75.6 | 60.8 | 86.3 | 72.5 |
| | EADA† | ✗ | 64.1 | 84.9 | 83.2 | 69.8 | 82.9 | 79.9 | 73.9 | 66.6 | 84.0 | 78.6 | 66.8 | 88.8 | 76.4 |
| | DUC | ✗ | 65.5 | 84.9 | 84.3 | 73.0 | 83.4 | 81.1 | 73.9 | 66.6 | 85.4 | 80.1 | 69.2 | 88.8 | 78.0 |
| SFADA | ELPT | ✓ | 65.3 | 84.1 | 84.9 | 72.9 | 84.4 | 82.8 | 69.8 | 63.3 | 86.1 | 76.2 | 65.6 | 89.1 | 77.0 |
| | DAPM-TT | ✓ | 64.4 | 85.8 | 85.4 | 72.4 | 84.7 | 84.1 | 70.0 | 63.3 | 85.6 | 77.4 | 65.8 | 89.1 | 77.3 |
| | MHPL† | ✓ | 65.8 | 86.0 | 85.5 | 72.9 | 86.5 | 84.7 | 73.0 | 65.0 | 86.2 | 77.7 | 68.3 | 89.2 | 78.4 |
| | LFTL | ✓ | 66.9 | 86.6 | 85.5 | 73.1 | 86.3 | 84.5 | 72.2 | 65.7 | 85.9 | 79.2 | 69.0 | 90.2 | 78.8 |
| | **ProULearn** | ✓ | 68.5 | 86.6 | 85.7 | 72.6 | 88.3 | 84.4 | 72.0 | 67.8 | 85.4 | 78.1 | 68.1 | 89.9 | **79.0** |

From the tables, it is shown that our proposed ProULearn method outperforms other benchmarks across various settings on all four datasets. When applied to larger-scale datasets such as VisDA-2017, Office-Home, and DomainNet-126, ProULearn achieves notable average performance gains of 1.5%, 0.6%, and 1.6% respectively over the MHLP method. On the smaller Office-31 dataset, our method has an average improvement of 0.5% compared to the second-best method, LFTL. A key advantage of ProULearn is its ability to achieve these performance gains while requiring labels for 5% of the target domain data before training, without needing access to source domain data. This significantly reduces the demand for human annotation and data storage compared to traditional

Table 2: Classification accuracy (%) on VisDA-2017 and Office-31 datasets with 5% labeled target domain samples.

| Categories | Method | SF | VisDA-2017 | Office-31 | | | | | | |
|---|---|---|---|---|---|---|---|---|---|---|
| | | | S→R | A→D | A→W | D→A | D→W | W→A | W→D | Avg |
| SFDA | SHOT | ✓ | 82.4 | 94.0 | 90.1 | 74.7 | 98.4 | 74.3 | 99.9 | 88.6 |
| | NRC | ✓ | 85.9 | 96.0 | 90.8 | 75.3 | 99.0 | 75.0 | 100.0 | 89.4 |
| | AaD | ✓ | 87.3 | 94.5 | 94.5 | 75.6 | 98.2 | 75.4 | 99.9 | 89.7 |
| | SF(DA)$^2$ | ✓ | 88.1 | 95.8 | 92.1 | 75.7 | 99.0 | 76.8 | 99.8 | 89.9 |
| Active DA | AADA | ✗ | 80.8 | 89.2 | 87.3 | 78.2 | 99.5 | 78.7 | 100.0 | 88.8 |
| | TQS | ✗ | 83.1 | 92.8 | 92.2 | 80.6 | 100.0 | 80.4 | 100.0 | 91.1 |
| | CLUE | ✗ | 83.3 | 92.0 | 87.3 | 79.0 | 99.2 | 79.6 | 99.8 | 89.5 |
| | EADA$^\dagger$ | ✗ | 86.5 | 96.8 | 96.7 | 82.7 | 100.0 | 81.3 | 100.0 | 92.9 |
| | DUC | ✗ | 88.9 | 95.8 | 96.4 | 81.9 | 99.6 | 81.4 | 100.0 | 92.5 |
| SFADA | ELPT | ✓ | 89.2 | 98.0 | 97.2 | 81.2 | 99.4 | 80.7 | 100.0 | 92.8 |
| | DAPM-TT | ✓ | 88.4 | 96.8 | 96.4 | 83.5 | 99.7 | 81.7 | 100.0 | 93.0 |
| | MHPL$^\dagger$ | ✓ | 90.3 | 98.2 | 96.6 | 81.4 | 99.0 | 82.1 | 100.0 | 92.9 |
| | LFTL | ✓ | **92.8** | 98.0 | 98.5 | 82.6 | 99.9 | 82.2 | 100.0 | 93.5 |
| | **ProULearn** | ✓ | 91.8 | 99.2 | 96.3 | 84.4 | 99.4 | 84.4 | 100.0 | **94.0** |

Table 3: Classification accuracy (%) of the DomainNet-126 dataset with 5% labeled target domain samples.

| Categories | Method | SF | R→C | R→P | S→R | P→C | S→C | P→S | C→S | S→P | R→S | P→R | C→P | C→R | Avg |
|---|---|---|---|---|---|---|---|---|---|---|---|---|---|---|---|
| SFDA | SHOT | ✓ | 68.7 | 67.8 | 76.3 | 63.9 | 71.5 | 57.4 | 59.9 | 65.5 | 57.7 | 78.8 | 62.0 | 78.0 | 67.3 |
| | AaD | ✓ | 69.3 | 68.6 | 77.4 | 65.3 | 72.9 | 61.3 | 59.4 | 67.5 | 57.1 | 79.9 | 62.5 | 78.7 | 68.3 |
| | PFC$^\dagger$ | ✓ | 71.9 | 70.3 | 80.4 | 72.7 | 76.7 | 67.6 | 62.3 | 68.5 | 61.0 | 83.1 | 65.1 | 77.7 | 71.4 |
| Active DA | CLUE | ✗ | 66.3 | 60.2 | 76.0 | 58.9 | 66.2 | 65.9 | 58.6 | 58.7 | 60.5 | 76.8 | 57.6 | 77.5 | 65.3 |
| | EADA | ✗ | 71.1 | 68.6 | 81.0 | 69.4 | 71.0 | 65.1 | 63.5 | 64.3 | 65.7 | 83.0 | 66.0 | 80.8 | 70.8 |
| | DUC | ✗ | 72.4 | 70.3 | 81.1 | 74.0 | 73.5 | 67.6 | 67.1 | 70.0 | 66.5 | 83.5 | 67.1 | 81.1 | 72.9 |
| SFADA | ELPT$^\dagger$ | ✓ | 64.3 | 64.7 | 83.7 | 66.6 | 59.0 | 64.1 | 57.1 | 61.0 | 56.4 | 83.7 | 65.5 | 84.1 | 67.5 |
| | DAPM-TT$^\dagger$ | ✓ | 73.0 | 74.5 | 84.8 | 72.1 | 74.3 | 66.6 | 65.9 | 71.4 | 67.1 | 85.9 | 70.4 | 84.6 | 74.2 |
| | MHPL$^\dagger$ | ✓ | 77.8 | 75.7 | 87.3 | 76.9 | 78.2 | 70.2 | 70.4 | 73.6 | 69.9 | 87.7 | 71.1 | 85.2 | 77.0 |
| | **ProULearn** | ✓ | 79.5 | 77.6 | 86.9 | 78.9 | 80.1 | 72.1 | 72.9 | 75.7 | 71.6 | 89.1 | 73.2 | 85.9 | **78.6** |

SFDA and active DA methods. The performance improvements across four diverse benchmark datasets demonstrate the effectiveness of our approach in various domain adaptation scenarios.

## 5 ABLATION STUDIES

In this section, a series of ablation studies are carried out to validate the proposed ProULearn method including loss component analysis, hyper-parameter sensitivity analysis and evaluation of the HPE strategy. More ablation studies including the influences of different budgets, validation of distribution measurement strategies, and analysis of feature distribution are presented in the Appendix.

### 5.1 HYPER-PARAMETER SENSITIVITY ANALYSIS

As described in Section 3, our method incorporates several hyper-parameters to facilitate the selection of active samples, including the number of trees $g$ in the HPE mechanism and the number of neighbors $K$ considered for each sample in the correlation index calculation. To better understand how to determine the optimal hyper-parameters, we conducted extensive experiments on the Office-31 dataset, varying $g$ within the range [100, 400] and $K$ within [4, 16]. The average performances of Office-31 under different hyper-parameter combinations are recorded in Figure 4a.

As the figure shows, the maximum average performance occurs when $g$ is 200 and $K$ is 8 for the Office-31 dataset. We applied a similar strategy to other datasets to determine their optimal parameters. We found that setting the tree number $g$ to 200 works well across all datasets, as this parameter is more related to the feature output dimension rather than specific dataset properties. Our empirical studies reveal that the number of trees $g$ plays a crucial role of the HPE mechanism. When $g$ is too low, the estimation becomes unstable since the selection of sample and feature is random, leading to unreliable selection. Conversely, when $g$ is too high, it may overfit to noise in the data.

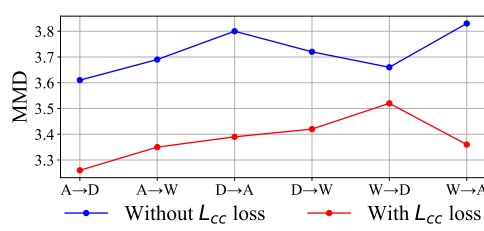

(a) Hyper-parameters sensitivity analysis.

(b) Comparison of maximum mean discrepancy with/without the presence of $L_{cc}$ during target domain adaptation on the Office-31 dataset.

Figure 4: Ablation studies on hyper-parameters (4a) and central correlation loss component (4b).

## 5.2 EVALUATION OF DIFFERENT CLUSTERING METHODS AND THE LOSS COMPONENT.

To evaluate the effectiveness of our proposed homogeneity propensity estimation (HPE) mechanism and the central correlation loss, we conducted comparative experiments using alternative traditional clustering methods and assessed the impact of the loss component. Figure 4b illustrates the maximum mean discrepancy (MMD) between the source and target domains for each transfer task in the Office-31 dataset, with and without the central correlation loss ($L_{cc}$). The consistently lower MMD values observed when $L_{cc}$ is applied demonstrate its efficacy in reducing domain discrepancy. This reduction in MMD indicates that our central correlation loss effectively encourages the model to learn more transferable features, thereby facilitating better domain adaptation.

Table 4: Ablation study on the Office-31 with different active sample selection strategies.

| Method | A→D | A→W | D→A | D→W | W→A | W→D | Avg. |
|---|---|---|---|---|---|---|---|
| ProULearn+*HDBSCAN* | 96.8 | 97.2 | 83.5 | 98.4 | 81.4 | 99.6 | 92.8 |
| ProULearn+*K-means* | 97.0 | 96.9 | 81.0 | 98.9 | 81.5 | 99.6 | 92.4 |
| ProULearn+*Entropy* | 95.4 | 95.9 | 79.2 | 97.1 | 80.4 | 100.0 | 91.3 |
| ProULearn+*PageRank* | 94.6 | 94.5 | 80.3 | 97.6 | 81.4 | 100.0 | 91.4 |
| **ProULearn+HPE** | 99.2 | 96.3 | 84.4 | 99.4 | 84.4 | 100.0 | **94.0** |

To further validate the superiority of our HPE mechanism, we compared it with three popular clustering methods: HDBSCAN (Campello et al., 2013), K-means (Lloyd, 1982), and PageRank (Page et al., 1999). In addition, we utilize commonly used feature entropy to calculate the importance score of each sample based on model prediction uncertainty. Table 4 presents the results of this comparison on the Office-31 dataset. The ProULearn framework consistently outperforms variants using other sample selection strategies, with an average accuracy improvement of 1.2% over the second-best method HDBSCAN. This performance gap underscores the advantages of our HPE approach in capturing the underlying structure of the target domain data. Unlike traditional clustering methods that rely on predefined distance metrics or density thresholds, HPE adapts to the local and global density of the data, making it more robust to diverse domain shift scenarios.

## 6 CONCLUSION

In this paper, we presented ProULearn, a novel Propensity-driven Uncertainty Learning framework for source-free active domain adaptation (SFADA). Our approach addresses the critical challenges in SFADA through innovative techniques for sample selection and model adaptation, comprising a homogeneity propensity estimation mechanism and correlation relationship learning. ProULearn effectively identifies informative samples while mitigating the impact of noisy outliers, and locates these samples before training begins, eliminating the need for acquiring human labels during the adaptation process. This proactive approach, combined with our training strategy that leads to more compact and discriminative class distributions in the target domain, significantly enhances the practicality of domain adaptation. Experiments across four benchmark datasets demonstrate ProULearn's consistent superiority over state-of-the-art methods in various domain adaptation scenarios.

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

APPENDIX

# A    RELATED WORKS ON SOURCE-FREE DOMAIN ADAPTATION

Source-free domain adaptation (SFDA) has been of great interest to researchers in recent years. It aims to adapt a pre-trained model on the source domain to other domains without access to source domain data. People apply this concept to many downstream tasks to solve real-world problems, for example, fine-grained image classification (Wang et al., 2020; Pan et al., 2024a), object detection (Li et al., 2021; VS et al., 2023), video analysis (Xu et al., 2022; Li et al., 2023a), etc. A large proportion of SFDA methods transfer the source domain knowledge to the target domains by exploring data properties (Li et al., 2024) like utilizing pseudo-labels (Liang et al., 2020), entropy minimization, and contrastive learning strategies. Xia et al. (2021) proposed an adaptive adversarial network (A$^2$Net) which utilized data augmentation and contrastive concepts to facilitate the classification in the target domains. Shen et al. (2023) introduced a selective pseudo-labeling and feature alignment strategy for domain adaptation. Yi et al. (2023) tackled the SFDA problem by learning with noise labels and leveraged early-time training phenomenon to address the label noise problem.

# B    ALGORITHMS

## B.1    TABLE OF NOTATIONS

| Notation | Description |
| --- | --- |
| $\mathcal{S}, \mathcal{X}_s, \mathcal{X}_s$ | Dataset, sample, and label sets that belong to the source domain. |
| $\mathcal{T}, \mathcal{X}_t, \mathcal{X}_t$ | Dataset, sample, and label sets that belong to the target domain. |
| $\mathcal{T}_l, \mathcal{T}_u$ | Selected active samples and unselected samples for the target domain. |
| $g$ | Number of trees for homogeneity propensity estimation (HPE). |
| $B$ | Budget (ratio) for active samples. |
| $v$ | Split value for a node (feature) in the tree. |
| $r(x)$ | Path length of sample $x$ in a tree. |
| $h(x)$ | Average path length of sample $x$ in all the trees. |
| $D$ | Dimension (length) of a feature vector. |
| $\bar{f}(x)$ | Mean of feature for sample x. |
| $C(f(x_i), f(x_j))$ | Correlation index for features of samples $x^i$ and $x^j$. |
| $K$ | Number of nearest neighbors. |
| $E(x)$ | Entropy (uncertainty) measurement in the class predictions for sample $x$. |
| $p(x)$ | Probability output for sample $x$. |
| $U$ | Sample selection score. |
| $o_c$ | Class centroid of class c. |
| $\delta_j(p(x))$ | $j^{th}$ logit output $p(x_i)$ of sample $x$. |
| $f(x)$ | Feature vector of sample $x$. |
| $z_{i,c}$ | Similarity score between sample $x_i$ and centroid of class c. |
| $M$ | Number of classes. |
| $\hat{y}_{tu}$ | Pseudo-labels for unselected samples. |
| $w(x)$ | Weight contribution to sample $x$ when calculating cross-entropy loss. |
| $N$ | Number of samples in the batch. |

Table 5: Summary of important notations used in this paper.

## B.2    HOMOGENEITY PROPENSITY ESTIMATION

The homogeneity score calculation process via HPE is given in Algorithm 1.

## B.3    OVERALL TRAINING PROCEDURE

The overall adaptation process of the proposed ProULearn framework is presented in Algorithm 2.

---

**Algorithm 1** Homogeneity Propensity Estimation (HPE) Procedures

---

**Require:** $\mathcal{X}_t = \{x_1, ..., x_n\}$: Set of target samples; $g$: Number of trees; $r$: Maximum tree depth.

1: **for** $i = 1$ to $g$ **do**
2:     Randomly select subset $\mathcal{X}_{subset}$ from $\mathcal{X}_t$.
3:     Build tree $T_i$ from $\mathcal{X}_{subset}$ with max depth $r$.
4: **end for**
5: **for** each $x \in \mathcal{X}_t$ **do**
6:     **for** $i = 1$ to $g$ **do**
7:         Compute path length $r_i(x)$ in tree $T_i$.
8:     **end for**
9:     Calculate homogeneity score $h(x)$ via Eq. 1.
10: **end for**
11: **return** $h(x)$ for each $x \in \mathcal{X}_t$.

---

This framework is an organized streamlined approach consisting of the following steps.

1. **Step 1:** Locate informative samples in the target domain via proposed homogeneity propensity estimation (HPE) and correlation index (CI) between samples.

2. **Step 2:** Assign pseudo-labels to the remaining unlabeled samples utilizing the homogeneity score from HPE and CI between samples and class centroids.

3. **Step 3:** Adapt the pre-trained source domain model to the target domain and keep refining pseudo-labels via loss components.

---

**Algorithm 2** ProULearn for Source-Free Active Domain Adaptation

---

**Require:** $\mathcal{X}_t, f_s$: Unlabeled target domain data; pre-trained source model.
**Require:** $B, K, g$: Labeling budget; number of nearest neighbors; number of Trees for HPE.

1: Compute homogeneity scores $h(x)$ for all samples in $\mathcal{X}_t$ via Eq. 1.
2: Calculate correlation index $C(x^i, x^j)$ for all sample pairs via Eq. 2.
3: Compute entropy $E_i$ for each sample and its neighbors via Eq. 3.
4: Obtain selection score combining $h(x_i)$ and $E_i$ by Eq. 4.
5: Select top $B\%$ samples with highest $U_i$ scores for labeling.
6: Initialize class centroids $o_c$ via Eq. 5.
7: Assign pseudo-labels to remaining unlabeled samples via Eq. 6.
8: **for** $e$ in $\{1, ..., \text{epochs}\}$ **do**
9:     **for** each mini-batch **do**
10:         Compute weighted cross-entropy loss, information maximization loss, and central correlation loss via Eq. 7 to Eq. 9.
11:         Calculate total loss $\mathcal{L}$ via Eq. 10.
12:         Update model parameters using SGD.
13:     **end for**
14:     **if** $e \bmod (\text{epochs}/10) == 0$ **then**
15:         Update class centroids $o_c$ via Eq. 5.
16:         Refine pseudo-labels for unlabeled samples via Eq. 6.
17:     **end if**
18: **end for**

---

The ProULearn algorithm addresses several key challenges in source-free active domain adaptation through its innovative approach to sample selection and model adaptation. By utilizing the HPE mechanism combined with correlation index calculation, ProULearn strikes a balance between se-

lecting informative samples and avoiding noisy outliers, which is crucial for effective domain adaptation. Unlike methods that rely solely on model uncertainty or traditional clustering techniques, ProULearn's sample selection strategy adapts to the global and local density of the data, making it more robust to diverse domain shifts.

### B.4 Discussions

The proposed ProULearn framework represents a significant advancement in SFADA through several key innovations. First, our HPE mechanism, combined with correlation-based feature relationships, introduces a novel approach to sample selection that considers both global data structure and local feature distributions, enabling more reliable identification of informative samples and pseudo-label assignment. Unlike traditional methods that rely heavily on initial distributions or require progressive labeling during training, ProULearn performs one-time sample selection before the adaptation process, making it particularly practical for real-world applications where continuous human annotation is infeasible. Second, our correlation index fundamentally differs from conventional distance metrics by capturing feature-level distribution relationships rather than spatial proximity, making it more robust to domain shifts. This correlation-based approach, when integrated with HPE, has demonstrated superior performance compared to traditional clustering methods. Furthermore, the newly proposed central correlation loss and weighted cross-entropy loss incorporate these components, creating more compact and discriminative class distributions. This cohesive integration of novel components forms a comprehensive framework tailored for SFADA challenges, while the underlying principles of our sample selection strategy offer broader applicability for various computer vision tasks where identifying informative samples or features is crucial.

## C Implementation Details

For the experiments, we employ the standard SFADA settings as in (Li et al., 2022; Wang et al., 2023; Du & Li, 2023). ResNet-50 (He et al., 2016) serves as the backbone for Office-31, Office-Home, and DomainNet-126, while ResNet-101 is utilized for VisDA-2017. All backbones are initialized with ImageNet-1K pre-trained weights. The hyperparameter $g$ is set to 200 across all datasets as it primarily depends on feature dimensionality, while $K$ is tuned based on dataset characteristics: smaller values ($K$=8) for Office-31 and Office-Home where classes have fewer samples, and larger values for DomainNet-126 ($K$=40) and VisDA-2017 ($K$=84). These values were determined through extensive experimentation as in the ablation studies (Section 5 in the paper) to optimize performance. Experiments are conducted on RTX A5000 GPUs using the PyTorch library (Paszke et al., 2019).

We train the model on the source domain for 100 epochs on all datasets. During target domain adaptation, the fine-tuning iteration is set to 30 epochs. The backbone and classifier's learning rate is set to $1e^{-3}$ for the VisDA-2017 dataset and $1e^{-2}$ for the other dataset as in (Liang et al., 2022; Xia et al., 2021; Wang et al., 2023). The bottleneck layer's learning rate is 10 times higher than the backbone. Optimization is performed using SGD with a 0.9 momentum. We conduct experiments mainly based on a 5% budget for active sampling, the same as other SFADA benchmarks. Batch sizes are set to 64 for Office-31 and 128 for other datasets. In addition, we also conduct experiments with 1% as well as 10% budget and the results are presented in Appendix Section E.1.

## D Detail results of VisDA-2017

Table 6 listed the benchmark performance on the VisDA-2017 dataset with detailed categorical results.

## E Additional ablation study results

### E.1 Influence of different budgets

We conduct experiments to evaluate our ProULearn framework's effectiveness under different labeling budgets (1% and 10%) across multiple datasets. With 10% labeled target samples on the Office-31 dataset (Table 7), ProULearn achieves strong performance with 95.6% average accuracy.

Table 6: Detail results of the VisDA-2017 dataset with 5% labeled target domain samples.

| Categories | Method | SF | plane | bcycl | bus | car | horse | knife | mcycl | person | plant | sktbrd | train | truck | Avg |
|---|---|---|---|---|---|---|---|---|---|---|---|---|---|---|---|
| SFDA | SHOT | ✓ | 94.6 | 87.5 | 80.4 | 59.5 | 92.9 | 95.1 | 83.1 | 80.2 | 90.9 | 89.2 | 85.8 | 56.9 | 83.0 |
| | NRC | ✓ | 96.1 | 90.8 | 83.9 | 61.5 | 95.7 | 95.7 | 84.4 | 80.7 | 94.0 | 91.9 | 89.0 | 59.5 | 85.3 |
| | AaD | ✓ | 96.8 | 89.3 | 83.8 | 82.8 | 96.5 | 95.2 | 90.0 | 81.0 | 95.7 | 92.9 | 88.9 | 54.6 | 87.3 |
| | SF(DA)$^2$ | ✓ | 96.8 | 89.3 | 82.9 | 81.4 | 96.8 | 95.7 | 90.4 | 81.3 | 95.5 | 93.7 | 88.5 | 64.7 | 88.1 |
| Active DA | AADA | ✗ | 85.7 | 80.0 | 81.3 | 81.9 | 95.6 | 81.3 | 85.2 | 81.7 | 82.6 | 80.6 | 80.1 | 54.1 | 80.8 |
| | TQS | ✗ | 86.4 | 83.2 | 86.7 | 83.5 | 93.3 | 86.2 | 88.8 | 78.1 | 88.9 | 84.7 | 81.3 | 56.2 | 83.1 |
| | CLUE | ✗ | 95.3 | 76.4 | 87.5 | 74.6 | 94.5 | 76.0 | 92.9 | 88.4 | 93.7 | 87.0 | 85.7 | 47.2 | 83.3 |
| SFADA | MHPL$^\dagger$ | ✓ | 96.9 | 91.6 | 89.6 | 83.5 | 97.1 | 96.6 | 91.2 | 88.7 | 94.1 | 93.5 | 91.1 | 70.2 | 90.3 |
| | **ProULearn** | ✓ | 97.2 | 93.7 | 88.9 | 83.6 | 97.4 | 97.2 | 90.9 | 91.9 | 97.1 | 96.1 | 92.5 | 74.5 | **91.8** |

On the larger-scale DomainNet-126 dataset with the same 10% budget (Table 8), our method maintains superior performance at 82.7% average accuracy. In addition, when examining low-budget scenarios using only 1% labeled samples on the VisDA-2017 dataset (Table 9), ProULearn still demonstrates strong adaptation capability with 87.6% average accuracy. These consistent performances across various datasets and budget settings validate that our method can effectively identify and leverage informative samples regardless of the labeling budget size.

Table 7: Classification accuracy (%) on Office-31 datasets with 10% labeled target domain samples.

| Categories | Method | SF | Office-31 | | | | | |
|---|---|---|---|---|---|---|---|---|
| | | | A→D | A→W | D→A | D→W | W→A | W→D | Avg |
| Active DA | AADA | ✗ | 93.5 | 93.1 | 83.2 | 99.7 | 84.2 | 100.0 | 92.3 |
| | TQS | ✗ | 96.4 | 96.4 | 86.4 | 100.0 | 87.1 | 100.0 | 94.4 |
| | CLUE | ✗ | 96.2 | 94.7 | 84.4 | 99.4 | 81.0 | 100.0 | 92.6 |
| | LADA | ✗ | 97.8 | 99.1 | 87.3 | 99.9 | 87.6 | 99.7 | 95.2 |
| SFADA | MHPL$^\dagger$ | ✓ | 98.8 | 96.7 | 85.2 | 99.1 | 86.7 | 100.0 | 94.4 |
| | LFTL | ✓ | 98.9 | 99.4 | 87.8 | 100.0 | 86.3 | 100.0 | 95.4 |
| | **ProULearn** | ✓ | 99.4 | 98.2 | 87.7 | 99.8 | 88.4 | 100.0 | **95.6** |

Table 8: Classification accuracy (%) of the DomainNet-126 dataset with 10% labeled target domain samples.

| Categories | Method | SF | R→C | R→P | S→R | P→C | S→C | P→S | C→S | S→P | R→S | P→R | C→P | C→R | Avg |
|---|---|---|---|---|---|---|---|---|---|---|---|---|---|---|---|
| SFADA | MHPL$^\dagger$ | ✓ | 82.2 | 79.5 | 90.3 | 82.2 | 83.7 | 77.1 | 76.2 | 78.5 | 75.1 | 90.9 | 77.4 | 90.2 | 81.9 |
| | **ProULearn** | ✓ | 83.0 | 80.4 | 90.1 | 82.7 | 84.5 | 77.8 | 77.3 | 79.8 | 76.5 | 90.8 | 78.9 | 90.0 | **82.7** |

Table 9: Detail results of the VisDA-2017 dataset with 1% labeled target domain samples.

| Categories | Method | SF | plane | bcycl | bus | car | horse | knife | mcycl | person | plant | sktbrd | train | truck | Avg |
|---|---|---|---|---|---|---|---|---|---|---|---|---|---|---|---|
| SFADA | MHPL$^\dagger$ | ✓ | 95.1 | 86.8 | 83.7 | 62.7 | 93.0 | 20.1 | 84.3 | 80.8 | 85.5 | 90.5 | 88.3 | 61.2 | 77.7 |
| | LFTL | ✓ | 95.9 | 84.6 | 84.6 | 77.1 | 95.4 | 93.6 | 91.4 | 87.1 | 93.2 | 90.4 | 87.8 | 67.6 | 87.4 |
| | **ProULearn** | ✓ | 95.3 | 89.6 | 86.1 | 72.3 | 95.3 | 95.5 | 89.5 | 84.9 | 93.5 | 93.0 | 88.6 | 67.0 | **87.6** |

## E.2 EVALUATION OF DIFFERENT DISTRIBUTION MEASUREMENT STRATEGIES.

To validate the proposed correlation index distribution measurement strategy, we conduct ablation studies comparing our correlation index with Euclidean distance and cosine similarity metrics. As demonstrated in Table 10, our correlation index-based approach shows superior performance with 94.0% average accuracy.

This improvement can be attributed to the correlation index's ability to capture the relationship between samples and their distribution patterns in the feature space, which is crucial for both the sample selection process and pseudo-label refinement. Unlike Euclidean distance which focuses on spatial proximity or cosine similarity which only considers angular relationships, the correlation in-

Table 10: Ablation study on the Office-31 dataset with different distribution measurement strategies.

| Method | A→D | A→W | D→A | D→W | W→A | W→D | Avg. |
|---|---|---|---|---|---|---|---|
| ProULearn+*Euclidian distance* | 96.8 | 97.2 | 83.5 | 98.4 | 81.4 | 99.6 | 92.8 |
| ProULearn+*Cosine similarity* | 97.0 | 96.9 | 81.0 | 98.9 | 81.5 | 99.6 | 92.4 |
| **ProULearn+*Correlation index*** | 99.2 | 96.3 | 84.4 | 99.4 | 84.4 | 100.0 | **94.0** |

dex evaluates feature-level distributions and better aligns with our proposed homogeneity propensity estimation mechanism, leading to more effective target domain adaptation.

### E.3 EMPIRICAL ANALYSIS OF FEATURE DISTRIBUTION

To validate the effectiveness of our ProULearn method in clustering samples and creating compact class distributions, we conducted an empirical analysis using the maximum mean discrepancy (MMD). We compared ProULearn with MHPL on the Office-31 dataset. Figure 5 illustrates the results of this analysis.

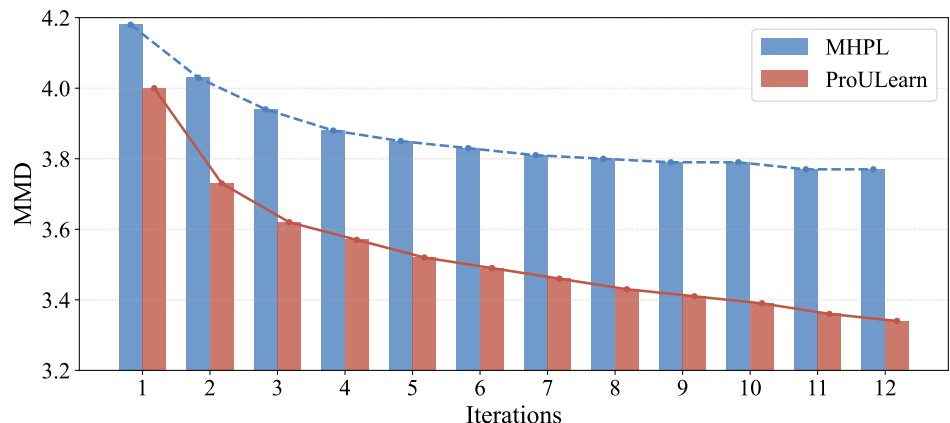

Figure 5: Maximum mean discrepancy (MMD) comparison between ProULearn and MHPL during training on the Office-31 dataset (A→D).

The MMD quantifies the dissimilarity between sample distributions and their respective class centroids. A lower MMD value indicates that samples are more tightly clustered around their class centroids. As evident from Figure 5, ProULearn consistently maintains lower MMD values throughout the training process compared to MHPL. This observation supports our claim that ProULearn achieves superior sample clustering and more compact class distributions. The lower MMD values indicate that samples are closer to their respective class centroids, which is crucial for accurate pseudo-label generation and overall adaptation performance. Furthermore, ProULearn has a steeper decline in MMD values during training. This rapid decrease suggests that by utilizing the proposed $L_{cc}$ loss and strategic active sample selection, our model can faster adapt to the target domain. The accelerated convergence to lower MMD values demonstrates ProULearn's ability to quickly identify and leverage discriminative features in the target domain.

