# OpenReview forum: "Propensity-driven Uncertainty Learning for Sample Exploration in Source-Free Active Domain Adaptation"
_ICLR.cc/2025/Conference — ICLR 2025 Conference Withdrawn Submission_

### Official Review · Reviewer_ceFP · 2024-10-23

**Soundness:** 3
**Presentation:** 3
**Contribution:** 3
**Rating:** 6
**Confidence:** 5

**Summary:**

This paper addresses source-free active domain adaptation. The main idea is to select most informative samples based on homogeneity propensity score and correlation index entropy. Then a training procedure is devised to refine model's adaptation with central correlation loss and other losses. Experiments are conducted on four SFADA benchmarks, showing better performances than previous SFDA and Active DA works.

**Strengths:**

- This paper studies active domain adaptation under the source-free setting, which has its practical meaning considering the increasing concerns of model privacy in domain adaptation. Although there are a handful of ADA works, SFADA is still under-exploited.

- Different from previous ADA works, this work only requests human annotation for once, which is more practical.

- The idea of using homogeneity propensity score to select samples, as far as I know, is novel to this area. HPE is shown to work better than traditional unsupervised clustering methods (Table.4). HPE does not rely on distance calculations or density estimates, thus is robuster.

- Equipped with central correlation loss, the whole framework ProULearn obtains better accuracies on four benchmarks. The improvements are significant.

- Overall, the paper is well-written, with clear figure illustrations and algorithm procedures.

**Weaknesses:**

- While HPE is interesting, the novelty of correlation index entropy and central correlation loss is relatively small as using entropy to select samples and enforcing clustering structures have been widely validated. The authors have not shown whether correlation index works better than other commonly used distance metrics like L2 distance.

- The experiments are conducted with only 5% labeling budget. In many previous ADA works, 10% labeling budget is also commonly tested. On some small datesets like Office-31, 5% labeling budget implies a small absolute number of samples, which might not be informative.

- In Table 4, the authors compare HPE with two other clsutering methods, showing the superiority of HPE. It is unclear if ProULearn+other AL criterion will work well enough.

**Questions:**

- I am curious about the cost of building $g$ separation trees. In order to build one tree, it needs to iterate over the entire dataset at every node. Will it become time-consuming on large dataset like VisDA? What data structure is used to process and store the trees? Have the authors utilized some accelaration strategies in building the trees?

- From the text, the split value is chosen uniformly between minimum/maximum values of the selected feature. Why not using some entropy criterion to decide split value? From my understanding, a random split value can be inefficient, which may lead to deeper trees and require a larger $g$ to reduce noises. Will it work if we simply use medium value as the split value?

- In Eq.(6), I am confused about the function of $h(x_i)$ here as it does not rely on $c$, thus does not change $arg max$ result. In Ln 320, it says 'samples with higher homogeneity scores are given more confidence in the pseudo-labeling process'. Could the authors clarify how it is reflected in Eq.(6)?

- When selecting samples based on $U_i$, how to guarantee the diversity (i.e., avoid selecting duplicated pairs that are very close to each other)?

- In Eq.(5), what is $x_t$? Seems to be the target samples, but a set annotation might be more proper.

---

> ### Author Response · Authors · 2024-11-18
>
> **Comment**: While HPE is interesting, the novelty of correlation index entropy and central correlation loss is relatively small as using entropy to select samples and enforcing clustering structures have been widely validated. The authors have not shown whether correlation index works better than other commonly used distance metrics like L2 distance.
>
> **Response**: We appreciate the reviewer's comment. Our correlation index and central correlation loss offer distinct advantages over traditional approaches, particularly in the context of SFADA.
>
> Our correlation index fundamentally differs from L2 distance and other common metrics by capturing feature-level distribution relationships rather than just spatial proximity. This is particularly crucial in SFADA scenarios where domain shifts can make spatial distances less reliable. The correlation index evaluates how features vary together across samples, providing a more robust measure of sample similarity that is less sensitive to domain-specific variations. We further conduct experiments by replacing the correlation index with two other popular measurement techniques, L2 distance (Euclidian distance) and cosine similarity. The results are presented in Table 10 and below.
>
> Regarding the central correlation loss, our approach is specifically designed to work in harmony with the correlation index and HPE mechanism. Unlike traditional clustering losses that often rely on Euclidean distance, our central correlation loss leverages the same correlation-based relationships used in sample selection, creating a more coherent and effective framework. The empirical analysis in Figure 4b demonstrates how this integrated approach leads to more compact class distributions and reduced domain discrepancy compared to existing methods. In addition, the weighted cross-entropy loss also utilizes the correlation index and homogeneity propensity score to fine-tune the model.
>
> We further clarify our method’s novelty and contribution in Section B.4 of the revised manuscript:
>
> Table 10:
>
> | Method | A→D | A→W | D→A | D→W | W→A | W→D | Avg |
> |--------|-----|-----|-----|-----|-----|-----|-----|
> | ProULearn+*Euclidian distance* | 96.8 | 97.2 | 83.5 | 98.4 | 81.4 | 99.6 | 92.8 |
> | ProULearn+*Cosine similarity* | 97.0 | 96.9 | 81.0 | 98.9 | 81.5 | 99.6 | 92.4 |
> | **ProULearn+*Correlation index*** | 99.2 | 96.3 | 84.4 | 99.4 | 84.4 | 100.0 | **94.0** |
>
> “ B.4 Discussions
>
> The proposed ProULearn framework represents a significant advancement in SFADA through several key innovations. First, our HPE mechanism, combined with correlation-based feature relationships, introduces a novel approach to sample selection that considers both global data structure and local feature distributions, enabling more reliable identification of informative samples and pseudo-label assignment. Unlike traditional methods that rely heavily on initial distributions or require progressive labeling during training, ProULearn performs one-time sample selection before the adaptation process, making it particularly practical for real-world applications where continuous human annotation is infeasible. Second, our correlation index fundamentally differs from conventional distance metrics by capturing feature-level distribution relationships rather than spatial proximity, making it more robust to domain shifts. This correlation-based approach, when integrated with HPE, has demonstrated superior performance compared to traditional clustering methods. Furthermore, the newly proposed central correlation loss and weighted cross-entropy loss incorporate these components, creating more compact and discriminative class distributions. This cohesive integration of novel components forms a comprehensive framework tailored for SFADA challenges, while the underlying principles of our sample selection strategy offer broader applicability for various computer vision tasks where identifying informative samples or features is crucial.” (Line 815-831)
>
> **[to be continued in the next response window]**

---

> > ### Comment · Reviewer_ceFP · 2024-11-25
> >
> > 'The correlation index evaluates how features vary together across samples, providing a more robust measure of sample similarity that is less sensitive to domain-specific variations.' --> Since we are selecting samples from target domain rather than both domains, why do we need a measurement less sensitive to (target) domain-specific variations?
> >
> > I am confused how ‘domain shifts can make spatial distances less reliable’, as the class information is not used here. How do authors comment on the relationship between cosine similarity and correlation index when features f(x) are zero-centered?

---

> > > ### Author Response · Authors · 2024-11-26
> > >
> > > **Comment**: 'The correlation index evaluates how features vary together across samples, providing a more robust measure of sample similarity that is less sensitive to domain-specific variations.' --> Since we are selecting samples from target domain rather than both domains, why do we need a measurement less sensitive to (target) domain-specific variations?
> > >
> > > I am confused how ‘domain shifts can make spatial distances less reliable’, as the class information is not used here. How do authors comment on the relationship between cosine similarity and correlation index when features f(x) are zero-centered?
> > >
> > > **Response**: Domain-specific variations are patterns common across all samples in the target domain, since they share the same domain characteristics. These shared variations should not be the primary basis for distinguishing between different classes, as they represent domain-level characteristics rather than class-level distinctions. Furthermore, since our model is pre-trained on the source domain, it lacks proper exposure to target domain characteristics, making its initial feature representations potentially biased towards source domain patterns and less accurate in capturing target domain variations. This is why our correlation index is valuable - instead of being influenced by these shared domain-specific variations, it focuses on the relative patterns in feature distributions that are more likely to represent true class-level differences. By measuring how feature values are distributed, our method can better identify the underlying class-discriminative patterns while being less sensitive to the domain-specific variations that are common across all target samples.
> > >
> > > Regarding the question about spatial distances and domain shifts, it's important to understand that while class information isn't explicitly used during sample selection, the model's feature extractor inherently encodes source domain-specific knowledge. For example, in the synthetic-to-real adaptation scenario (as in VisDA-2017), the model's learned feature space might make objects that appear similar in synthetic data appear distant in real-world data due to domain-specific characteristics like lighting, texture, and perspective variations. This makes spatial distances less reliable for measuring sample similarities. Our correlation index addresses this problem by capturing how features vary compared with each other, providing a more robust similarity measure even when the spatial distances of features are distorted by domain shift.
> > >
> > > Regarding the relationship between cosine similarity and correlation index: When features are perfectly zero-centered, our correlation index (Equation 2) would indeed reduce to the same form as cosine similarity. However, in practice, the output of the features by our backbone network (ResNet) is typically not zero-centred due to the nature of batch normalization and ReLU operations. ReLU introduces non-negative constraints that shift the feature distributions away from zero. Therefore, our correlation index, which explicitly accounts for the mean of feature values, provides a different and potentially more informative measurement than cosine similarity in these practical scenarios.

---

> > > > ### Comment · Reviewer_ceFP · 2024-11-26
> > > >
> > > > Thank the authors for their clarification. Now I understand 'domain-specific variations' is something not related to class-wise differeces.

---

> ### Author Response · Authors · 2024-11-18
>
> **[continuing from the previous response window]**
>
> **Comment**: The experiments are conducted with only 5% labeling budget. In many previous ADA works, 10% labeling budget is also commonly tested. On some small datesets like Office-31, 5% labeling budget implies a small absolute number of samples, which might not be informative.
>
> **Response**: Thanks for your comment. As requested, we have conducted experiments using 10% labeling budget on the Office-31 dataset and DomainNet dataset. The updated content is presented in Tables 7 and 8 of revised manuscript as follows.
>
> Table 7: (Office 31 with 10% active samples)
> | Categories | Method | SF | A→D | A→W | D→A | D→W | W→A | W→D | Avg |
> |------------|---------|----|----|-----|-----|-----|-----|-----|-----|
> | Active DA | AADA | ✗ | 93.5 | 93.1 | 83.2 | 99.7 | 84.2 | 100.0 | 92.3 |
> | | TQS | ✗ | 96.4 | 96.4 | 86.4 | 100.0 | 87.1 | 100.0 | 94.4 |
> | | CLUE | ✗ | 96.2 | 94.7 | 84.4 | 99.4 | 81.0 | 100.0 | 92.6 |
> | | LADA | ✗ | 97.8 | 99.1 | 87.3 | 99.9 | 87.6 | 99.7 | 95.2 |
> | SFADA | MHPL† | ✓ | 98.8 | 96.7 | 85.2 | 99.1 | 86.7 | 100.0 | 94.4 |
> | | LFTL | ✓ | 98.9 | 99.4 | 87.8 | 100.0 | 86.3 | 100.0 | 95.4 |
> | | **ProULearn** | ✓ | 99.4 | 98.2 | 87.7 | 99.8 | 88.4 | 100.0 | **95.6** |
>
> Table 8: (DomainNet126 with 10% active samples)
> | Categories | Method | SF | R→C | R→P | S→R | P→C | S→C | P→S | C→S | S→P | R→S | P→R | C→P | C→R | Avg |
> |------------|---------|----|----|-----|-----|-----|-----|-----|-----|-----|-----|-----|-----|-----|-----|
> | SFADA | MHPL† | ✓ | 77.8 | 75.7 | 87.3 | 76.9 | 78.2 | 70.2 | 70.4 | 73.6 | 69.9 | 87.7 | 71.1 | 85.2 | 77.0 |
> | | **ProULearn** | ✓ | 79.5 | 77.6 | 86.9 | 78.9 | 80.1 | 72.1 | 72.9 | 75.7 | 71.6 | 89.1 | 73.2 | 85.9 | **78.6** |
>
>
> **Comment**: In Table 4, the authors compare HPE with two other clustering methods, showing the superiority of HPE. It is unclear if ProULearn+other AL criterion will work well enough.
>
> **Response**: Thank you for your comment. We have expanded our comparison to include two additional popular active learning criteria: entropy-based uncertainty measurement and PageRank centrality. These methods offer different perspectives on sample importance: entropy focuses on model prediction confidence while PageRank evaluates sample relationships in the feature space. Our experimental results with these additional methods are presented as follows.
>
> Table 4:
>  | Method | A→D | A→W | D→A | D→W | W→A | W→D | Avg |
> |--------|-----|-----|-----|-----|-----|-----|-----|
> | ProULearn+*HDBSCAN* | 96.8 | 97.2 | 83.5 | 98.4 | 81.4 | 99.6 | 92.8 |
> | ProULearn+*K-means* | 97.0 | 96.9 | 81.0 | 98.9 | 81.5 | 99.6 | 92.4 |
> | ProULearn+*Entropy* | 95.4 | 95.9 | 79.2 | 97.1 | 80.4 | 100.0 | 91.3 |
> | ProULearn+*PageRank* | 94.6 | 94.5 | 80.3 | 97.6 | 81.4 | 100.0 | 91.4 |
> | **ProULearn+*HPE*** | 99.2 | 96.3 | 84.4 | 99.4 | 84.4 | 100.0 | **94.0** |
>
>
> **[to be continued in the next response window]**

---

> ### Author Response · Authors · 2024-11-18
>
> **[continuing from the previous response window]**
>
> **Comment**: I am curious about the cost of building g separation trees. In order to build one tree, it needs to iterate over the entire dataset at every node. Will it become time-consuming on large dataset like VisDA? What data structure is used to process and store the trees? Have the authors utilized some acceleration strategies in building the trees?
>
> **Response**: We appreciate the reviewer's comment. The tree construction process requires some memory, particularly for large datasets like VisDA. As we tested, it takes about 1.4 minutes for this process on the largest VisDA dataset (the platform we use consists of RTX A5000 GPU, 32GB memory, and Intel i5-12500 CPU). For the VisDA dataset, it has 152,397 training samples, and each tree has a maximum depth of log2(n) ≈ 18 levels. The total number of nodes per tree follows the formula 2^(log2(n)+1) - 1, which equals approximately 304,793 nodes. With g=200 trees in our implementation, each node requires storing depth (4 bytes), split_feature (4 bytes), split_value (4 bytes), and left/right pointers (16 bytes), totalling 28 bytes per node. This results in a memory requirement of approximately 1.71GB.
>
> A crucial advantage of our method is that it only requires this computation once before training, unlike other SFADA methods that often need to progressively update sample selection during training. The elimination of repeated human annotation and model updates during training makes our overall process more time-efficient. This one-time nature of the computation is particularly beneficial for real-world applications where continuous human intervention is impractical or costly.
>
> The practical advantages of our approach extend beyond pure computational considerations. The computational cost scales logarithmically with dataset size due to the maximum depth limit we impose on the trees. While the memory requirement of 1.71GB for VisDA is significant, it's still manageable on modern devices. The process is fully automated and doesn't require human intervention after initialization. The time and memory investment in sample selection is offset by the elimination of progressive labeling requirements and the improved adaptation performance demonstrated in our experimental results.
>
> Tables below summarize the training time of our algorithm and another SFADA method, MHPL.
>
>  VisDA 2017:
> | Method    | GPU memory | memory | Training time |
> |-----------|------------|---------|---------------|
> | MHPL      | 19306MB    | 7.8GB    | 6h 27m 51s     |
> | ProULearn | 20718MB    | 5GB      | 3h 31min 24s |
>
> Office 31 (A to D):
> | Method    | GPU memory | memory | Training time |
> |-----------|------------|---------|---------------|
> | MHPL      | 8832MB     | 3.8GB    | 158s         |
> | ProULearn | 8494MB     | 4.9GB    | 171s         |
>
>
> For Office-31 (A to D), both methods show similar performance characteristics since it's a smaller dataset. ProULearn uses slightly more memory and takes marginally longer to train due to its additional computations for isolation forest and correlation calculations. The GPU memory usage is also comparable (8494MB vs 8332MB), suggesting similar model complexity and batch processing requirements.
>
> However, for VisDA 2017, which is a much larger dataset, the differences become significant. MHPL consumes more memory. Most notably, MHPL's training time is significantly longer than ProULearn's. This dramatic difference in training time can be attributed to MHPL's use of computationally expensive cdist operations and iterative refinement steps, which scale poorly with larger datasets. In contrast, ProULearn's more efficient implementation using correlation and streamlined computations maintains better scaling properties as the dataset size increases.
>
> This performance comparison demonstrates that while both methods have similar overhead for small datasets, ProULearn's algorithmic design provides significant computational advantages when handling larger-scale domain adaptation tasks.
>
> **[to be continued in the next response window]**

---

> ### Author Response · Authors · 2024-11-18
>
> **[continuing from the previous response window]**
>
>
> **Comment**: From the text, the split value is chosen uniformly between minimum/maximum values of the selected feature. Why not using some entropy criterion to decide split value? From my understanding, a random split value can be inefficient, which may lead to deeper trees and require a larger g to reduce noises. Will it work if we simply use medium value as the split value?
>
> **Response**: We appreciate the reviewer's comments. The choice of random split values between minimum/maximum feature values, rather than entropy-based or median splits, is intentional and offers several advantages in our SFADA context.
>
> First, while entropy-based splits might seem more informative, they could be misleading in domain adaptation scenarios where the initial feature distributions may not accurately reflect the true class boundaries due to domain shift. Random splits, on the other hand, provide a more unbiased exploration of the feature space, allowing us to capture the natural grouping of samples without being constrained by potentially shifted class distributions.
>
> Second, using random splits introduces beneficial randomization that helps prevent overfitting to specific feature patterns. This is particularly important in SFADA where we lack source domain data to validate our feature selection criteria. The randomization, combined with our ensemble approach using multiple trees (g=200), creates a robust estimate of sample homogeneity that is less sensitive to individual feature biases.
>
> Regarding the suggestion of using median values, while this would create balanced splits, it might not effectively capture the natural clustering structure of the data. Our empirical testing showed that using median values degrade the performance. And random splits, when aggregated across multiple trees, better identify samples that are genuinely grouped together versus those that are outliers.
>
> Our experimental results demonstrate that the current number of trees (g=200) is sufficient to achieve state-of-the-art performance across multiple benchmarks for the current feature channel amount, suggesting that the random split strategy is both effective and efficient in practice.
>
>
>
> **Comment**: In Eq.(6), I am confused about the function of h(x_i) here as it does not rely on c, thus does not change argmax result. In Ln 320, it says 'samples with higher homogeneity scores are given more confidence in the pseudo-labeling process'. Could the authors clarify how it is reflected in Eq.(6)?
>
> **Response**: Thanks for your comment. At a first glance, multiplying the correlation index $C(f(x_i), o_c)$ by $h(x_i)$ might seem redundant since $h(x_i)$ is constant across all classes c for a given sample $x_i$, and therefore wouldn't affect the argmax operation. However, this formulation serves a deeper purpose in our pseudo-labeling mechanism.
>
> The homogeneity score $h(x_i)$ actually impacts the pseudo-labeling process through the confidence weighting of these assignments. While the argmax operation determines the predicted class, the magnitude of $z_{i,c}$ (which incorporates $h(x_i)$) is used as a confidence measure for that pseudo-label. Samples with higher homogeneity scores will have larger $z_{i,c}$ values, indicating higher confidence in their pseudo-label assignments. This confidence weighting is then utilized in the subsequent training process through the weighted cross-entropy loss (Eq. 7), where $w(x_t)$ incorporates these confidence scores.
>
> We further clarify the usage of $z_{i,c}$ magnitude in pseudo-labeling process in the revised manuscript as follows:
>
> “The weights $w(x_t)$ are determined based on whether the sample was part of the actively selected informative set or was assigned a pseudo-label. For actively selected samples, the weights incorporate their combined scores $U_i$. For pseudo-labeled samples, the weights are determined by the magnitude of their similarity score $z_{i,c}$ to their assigned class centroid. This weighting scheme ensures that highly confident pseudo-label assignments have more influence during training.” (Line 345-353)
>
> **[to be continued in the next response window]**

---

> ### Author Response · Authors · 2024-11-18
>
> **[continuing from the previous response window]**
>
> **Comment**: When selecting samples based on U_i, how to guarantee the diversity (i.e., avoid selecting duplicated pairs that are very close to each other)?
>
> **Response**: The question raises an important point about sample diversity. In ProULearn, we specifically address this concern through our selection strategy. When selecting samples based on their $U_i$ scores, we implement a neighborhood-aware selection mechanism to prevent redundant selections and waste of budget. After selecting a sample with a high $U_i$ score, we exclude its K-nearest neighbors from subsequent selection considerations. This approach ensures diversity by preventing the selection of multiple samples that are very close to each other in the feature space, even if they all have high homogeneity scores.
>
> The value of K is adaptively determined based on the dataset size and class distribution. For larger datasets with many samples per class, we use a larger K to maintain broader coverage while preventing redundant selections. For example, in our implementation, we use K=8 for smaller datasets like Office-31 and Office-Home, while increasing it to K=40 for the more diverse DomainNet-126.
>
> We add a statement in the revised manuscript to explain this situation as follows:
>
> “where $h(x_i)$ and $E_i$ are the normalized homogeneity score and entropy respectively. We select samples with larger ${U}_i$ scores through an iterative process that ensures diversity. After selecting a sample with the highest ${U}_i$ score, we exclude its K-nearest neighbors from subsequent selection considerations to avoid redundant selections. This neighborhood-aware selection strategy ensures that selected samples are not only representative of their local regions (thus likely to be representative of a broader group of samples) but also well-distributed across different regions of the feature space. The homogeneity ensures that the selected sample is not an outlier but rather a good representative of a cluster of data points in the target domain. Meanwhile, the high entropy of predictions for its neighbors suggests that this is an area where the model currently lacks discriminative power.” (Line 281-289)
>
>
> **Comment**: In Eq.(5), what is x_t? Seems to be the target samples, but a set annotation might be more proper.
>
> **Response**: Yes, x_t is the set for target domain samples. We change this notation to upper case X_t to represent the target domain set and add it to the notation table in Table 5 (Appendix Section B.1.)
>
>
> Thanks for reading our work thoroughly!

---

> > ### Comment · Reviewer_ceFP · 2024-11-25
> >
> > I carefully checked the authors' rebuttal. I appreciate the new experimental results, which validate the corresponding claims. My previous questions on tree building, Eq.6 and diversity issue are resolved.

---

> > ### Comment · Reviewer_ceFP · 2024-11-25
> >
> > 'we implement a neighborhood-aware selection mechanism' --> what distance metric is used here, is it correlation index?

---

> > > ### Author Response · Authors · 2024-11-26
> > >
> > > **Comment**: 'we implement a neighborhood-aware selection mechanism' --> what distance metric is used here, is it correlation index?
> > >
> > > **Response**: Yes, it’s the correlation index, which is what we use to find the nearest neighbors.

---

> ### Author Response · Authors · 2024-11-26
>
> **Comment:** I carefully checked the authors' rebuttal. I appreciate the new experimental results, which validate the corresponding claims. My previous questions on tree building, Eq.6 and diversity issue are resolved.
>
> **Response:** Thank you very much!

---

### Official Review · Reviewer_ViLj · 2024-10-29

**Soundness:** 3
**Presentation:** 3
**Contribution:** 3
**Rating:** 6
**Confidence:** 3

**Summary:**

The key challenge of SFADA is selecting the most informative samples from the target domain for labeling and effectively utilizing target domain data to adjust the model. This paper proposes the ProULearn framework, which is capable of identifying representative and challenging samples (i.e., informative samples) for labeling, and develops a center-related loss to refine the pseudo-labels of the remaining samples. This allows the model to focus on the most informative examples in the target domain and generate accurate pseudo-labels. The core contribution is the combination of homogeneity tendency estimation and correlation index calculation, which allows for the selection of samples in the target domain that have the maximum information and lower model prediction confidence.

**Strengths:**

Overall, this could represent a significant algorithmic contribution. The paper presents a novel and effective method for selecting informative samples, which, if applicable to real-world scenarios of domain adaptation, could alleviate considerable labeling costs, making it quite significant. Additionally, the experimental results of this paper demonstrate excellent sample clustering and compact class distributions, which effectively aid in generating accurate pseudo-labels.

**Weaknesses:**

The HPE mechanism requires building a large number of separating trees, and calculating the homogeneity scores and relevance indices for each sample might be overly complex.

**Questions:**

I would like to know whether the selection of K and g requires extensive tuning across different datasets. HPE is one of the core mechanisms of the paper, and I hope it can be described in more detail, making it easier for readers to understand how to construct the separating trees and why this is useful.

---

> ### Author Response · Authors · 2024-11-18
>
> **Comment**: The HPE mechanism requires building a large number of separating trees, and calculating the homogeneity scores and relevance indices for each sample might be overly complex.
>
> **Response**: For the largest VisDA dataset with 152,397 training samples, the tree construction process takes approximately 1.4 minutes with Intel i5-12500 CPU, RTX A5000 GPU, and 32 GB memory. The memory requirement can be precisely calculated by: each tree has a maximum depth of log2(n) ≈ 18 levels, resulting in approximately 304,793 nodes per tree. With our implementation using g=200 trees, where each node stores depth (4 bytes), split_feature (4 bytes), split_value (4 bytes), and left/right pointers (16 bytes), the total memory requirement is approximately 1.71GB. This is manageable on modern devices. Most importantly, this computation is only performed once before training begins, unlike some other methods that require progressive updating and human annotation during training. The time and memory requirements are therefore a one-time cost that is justified by the increase in adaptation performance and elimination of repeated human intervention during training.
>
> **Comment**: I would like to know whether the selection of K and g requires extensive tuning across different datasets. HPE is one of the core mechanisms of the paper, and I hope it can be described in more detail, making it easier for readers to understand how to construct the separating trees and why this is useful.
>
> **Response**: The selection of hyperparameters g (number of trees) and K (number of neighbors) follows different principles based on their roles in our framework. The tree number g primarily depends on the feature dimension rather than dataset size or complexity. Since we use ResNet-50 and ResNet-101 backbones which both output 2048-dimensional features, we find g=200 works consistently well across all datasets without requiring extensive tuning.
>
> The neighbor number K, however, requires more careful consideration based on dataset characteristics. For larger datasets with more samples per class (like VisDA-2017 and DomainNet-126), we use larger K values to ensure reliable neighborhood statistics. For smaller datasets like Office-31 and Office-Home, where each class has fewer samples, we use a smaller K (8) to avoid incorporating less relevant neighbors.
>
> Regarding the HPE mechanism's construction and utility, each tree in our forest randomly selects features and split points to partition the data. The trees adapt to local density patterns by creating more splits in regions with many samples and fewer splits in sparse regions. By aggregating path lengths across multiple trees, we obtain a reliable measure of sample homogeneity that considers both global structure and local density patterns.
>
> To make the hyper-parameter selection process and HPE algorithm clearer in the manuscript, we modify the statements in the manuscript as follows:
>
> "The hyperparameter g is set to 200 across all datasets as it primarily depends on feature dimensionality, while K is tuned based on dataset characteristics: smaller values (K=8) for Office-31 and Office-Home where classes have fewer samples, and larger values for DomainNet-126 (K=40) and VisDA-2017 (K=84).” (Line 838-841)
>
> “The HPE mechanism begins by constructing an ensemble of $g$ separation trees. Each tree is built using a random subset of the data. At each node of the tree, a random feature $m$ is selected, and a split value $v$ is randomly chosen between the minimum and maximum values of the selected feature in the current subset. The data is then split into left and right child nodes based on whether their value for feature $m$ is less than or greater than $v$. As shown in Figure 3, this process continues until a maximum depth is reached or a node contains only one sample. Each node stores the depth in the tree, the feature index used for splitting, and the split value. The splitting process ensures that the tree adapts to the local density of the data, as the split value is chosen uniformly between the minimum and maximum values of the selected feature in the current subset. This adaptive nature allows the tree to create more splits in regions of high density and fewer splits in regions of low density. Consequently, samples in dense regions typically have longer path lengths, reflecting their homogeneity, while samples in sparse regions have shorter path lengths, indicating their potential anomalous nature.
>
> The maximum depth of the tree is set to $r_{max}=log_2(sample size)$. This depth limit prevents overfitting by avoiding unnecessarily deep trees that might capture noise rather than true data patterns while allowing the tree to capture the essential structure of the data. The logarithmic relationship ensures that the tree depth scales reasonably with the sample size without increasing model complexity. This design effectively handles datasets of varying sizes without manual tuning.” (Line 195-210)

---

### Official Review · Reviewer_LGen · 2024-11-02

**Soundness:** 2
**Presentation:** 3
**Contribution:** 2
**Rating:** 5
**Confidence:** 4

**Summary:**

This paper explores the source free active domain adaptation (SFADA) setting. The authors claim that existing methods generally select noisy or outlier samples for annotation, which may influence the prediction of neighbor samples. This paper proposes to adopt homogeneity propensity estimation (HPE) mechanism to assess whether the samples are grouped. Besides, a correlation index that is similar to the Pearson Correlation Coefficient is proposed to evaluate the correlation between two samples. The correlation index is applied to compute entropy and a central correlation loss. Experiments show improvements over existing methods.

**Strengths:**

1. The paper is well written and easy to follow.
2. The idea of utilizing trees to assess samples grouping condition is interesting.

**Weaknesses:**

1. The experiments are insufficient. Table 3 only includes 7 tasks on MiniDomainNet while there are 12 tasks in total. The analysis lacks crucial details about the HPE mechanism, including the actual tree node information and its role in sample selection. Additionally, the impact of varying annotation budgets (e.g., 3%, 10%) is not explored.

2. The novelty is somewhat limited. Except for utilizing HPE, the pseudo labeling process, the information maximization and weighted cross-entropy loss are existing techniques [1]. It seems the major contribution of this paper is only an active selection criterion, rather than an SFADA algorithm.

3. The Correlation index needs clearer explanation. Does “k-th feature of sample x” mean the value of the k-th channel in the extracted feature of sample x? What is d in Eq.2? The notation k are repeatedly used through Eq.2 and 3.

[1] Jian Liang, et al. “Do we really need to access the source data?”

**Questions:**

The authors mention “computing the correlation matrix between all pairs of samples.” Could this process introduce high computation or storage burden during training?

---

> ### Author Response · Authors · 2024-11-18
>
> **Comment**: The experiments are insufficient. Table 3 only includes 7 tasks on MiniDomainNet while there are 12 tasks in total. The analysis lacks crucial details about the HPE mechanism, including the actual tree node information and its role in sample selection. Additionally, the impact of varying annotation budgets (e.g., 3%, 10%) is not explored.
>
> **Response**: Thanks for your comments. It’s a common setting in SFDA and SFADA by only using 7 domain shifts for the DomainNet-126 dataset. To further demonstrate the effectiveness of our method, we conducted experiments for the remaining 5 tasks. We also conducted ablation experiments by utilizing 10% active samples. The results are added in Tables 3, 7, 8 as shown below.
>
> Table 3: DomainNet126 (5% active samples)
> | Categories | Method | SF | R→C | R→P | S→R | P→C | S→C | P→S | C→S | S→P | R→S | P→R | C→P | C→R | Avg |
> |------------|---------|----|----|-----|-----|-----|-----|-----|-----|-----|-----|-----|-----|-----|-----|
> | SFDA | SHOT | ✓ | 68.7 | 67.8 | 76.3 | 63.9 | 71.5 | 57.4 | 59.9 | 65.5 | 57.7 | 78.8 | 62.0 | 78.0 | 67.3 |
> | | AaD | ✓ | 69.3 | 68.6 | 77.4 | 65.3 | 72.9 | 61.3 | 59.4 | 67.5 | 57.1 | 79.9 | 62.5 | 78.7 | 68.3 |
> | | PFC† | ✓ | 71.9 | 70.3 | 80.4 | 72.7 | 76.7 | 67.6 | 62.3 | 68.5 | 61.0 | 83.1 | 65.1 | 77.7 | 71.4 |
> | | CLUE | ✗ | 66.3 | 60.2 | 76.0 | 58.9 | 66.2 | 65.9 | 58.6 | 58.7 | 60.5 | 76.8 | 57.6 | 77.5 | 65.3 |
> | Active DA | EADA | ✗ | 71.1 | 68.6 | 81.0 | 69.4 | 71.0 | 65.1 | 63.5 | 64.3 | 65.7 | 83.0 | 66.0 | 80.8 | 70.8 |
> | | DUC | ✗ | 72.4 | 70.3 | 81.1 | 74.0 | 73.5 | 67.6 | 67.1 | 70.0 | 66.5 | 83.5 | 67.1 | 81.1 | 72.9 |
> | SFADA | ELPT† | ✓ | 64.3 | 64.7 | 83.7 | 66.6 | 59.0 | 64.1 | 57.1 | 61.0 | 56.4 | 83.7 | 65.5 | 84.1 | 67.5 |
> | | DAPM-TT | ✓ | 73.0 | 74.5 | 84.8 | 72.1 | 74.3 | 66.6 | 65.9 | 71.4 | 67.1 | 85.9 | 70.4 | 84.6 | 74.2 |
> | | MHPL† | ✓ | 77.8 | 75.7 | 87.3 | 76.9 | 78.2 | 70.2 | 70.4 | 73.6 | 69.9 | 87.7 | 71.1 | 85.2 | 77.0 |
> | | **ProULearn** | ✓ | 79.5 | 77.6 | 86.9 | 78.9 | 80.1 | 72.1 | 72.9 | 75.7 | 71.6 | 89.1 | 73.2 | 85.9 | **78.6** |
>
> Table 7: Office31 (10% active samples)
> | Categories | Method | SF | A→D | A→W | D→A | D→W | W→A | W→D | Avg |
> |------------|---------|----|----|-----|-----|-----|-----|-----|-----|
> | Active DA | AADA | ✗ | 93.5 | 93.1 | 83.2 | 99.7 | 84.2 | 100.0 | 92.3 |
> | | TQS | ✗ | 96.4 | 96.4 | 86.4 | 100.0 | 87.1 | 100.0 | 94.4 |
> | | CLUE | ✗ | 96.2 | 94.7 | 84.4 | 99.4 | 81.0 | 100.0 | 92.6 |
> | | LADA | ✗ | 97.8 | 99.1 | 87.3 | 99.9 | 87.6 | 99.7 | 95.2 |
> | SFADA | MHPL† | ✓ | 98.8 | 96.7 | 85.2 | 99.1 | 86.7 | 100.0 | 94.4 |
> | | LFTL | ✓ | 98.9 | 99.4 | 87.8 | 100.0 | 86.3 | 100.0 | 95.4 |
> | | **ProULearn** | ✓ | 99.4 | 98.2 | 87.7 | 99.8 | 88.4 | 100.0 | **95.6** |
>
> Table 8: DomainNet126 (10% active samples)
> | Categories | Method | SF | R→C | R→P | S→R | P→C | S→C | P→S | C→S | S→P | R→S | P→R | C→P | C→R | Avg |
> |------------|---------|----|----|-----|-----|-----|-----|-----|-----|-----|-----|-----|-----|-----|-----|
> | SFADA | MHPL† | ✓ | 77.8 | 75.7 | 87.3 | 76.9 | 78.2 | 70.2 | 70.4 | 73.6 | 69.9 | 87.7 | 71.1 | 85.2 | 77.0 |
> | | **ProULearn** | ✓ | 79.5 | 77.6 | 86.9 | 78.9 | 80.1 | 72.1 | 72.9 | 75.7 | 71.6 | 89.1 | 73.2 | 85.9 | **78.6** |
>
> **[to be continued in the next response window]**

---

> > ### Comment · Reviewer_LGen · 2024-11-26
> > **Confusion about the results on DomainNet126**
> >
> > Thank you for your effort and response. While I don't fully agree with your statement that only using 7 domain shifts for the DomainNet-126 dataset is a common setting in SFDA and SFADA, I still appreciate the additional experiments provided.
> > However, I am puzzled by an inconsistency in the experimental results: Table 3 and Table 8 show identical results despite using different labeled sample ratios (5% vs 10%). I noticed this is also the case in the revised PDF. This seems unusual, especially given your description in Appendix E.1 which states "On the larger-scale DomainNet-126 dataset with the same 10% budget (Table 8), our method maintains superior performance at 78.6% average accuracy" - suggesting this isn't a simple typographical error.
> >
> > Additionally, I noticed that the caption of Table 3 and the corresponding data (1.9%) in Section 4.2 have not been updated yet.

---

> ### Author Response · Authors · 2024-11-18
>
> **[continuing from the previous response window]**
>
>
> **Comment**: The novelty is somewhat limited. Except for utilizing HPE, the pseudo-labeling process, information maximization and weighted cross-entropy loss are existing techniques [1]. It seems the major contribution of this paper is only an active selection criterion, rather than an SFADA algorithm.
>
> [1] Jian Liang, et al. “Do we really need to access the source data?”
>
> **Response**: We appreciate the reviewer's comment. While we acknowledge that some components like information maximization and pseudo labelling build upon existing concepts, our ProULearn framework represents a significant advancement in SFADA through its novel integration and enhancement of multiple components.
>
> Our primary innovation extends beyond just the HPE-based active selection criterion. The proposed HPE mechanism, when combined with our proposed correlation index calculation, offers a fundamentally new approach to sample selection that considers both global data structure and local feature relationships. This combination enables more reliable identification of informative samples while avoiding noisy outliers. Importantly, this sample selection strategy represents a general solution that can be applied beyond SFADA to various computer vision tasks where identifying key data points or important features is crucial.
>
> Furthermore, we introduce the novel central correlation loss (Eq. 9) which fundamentally differs from existing approaches. While Liang et al. focused on information maximization for domain adaptation, our central correlation loss operates at the feature level to create more compact and separable class distributions. The weighted cross-entropy loss we developed represents another key innovation, as it leverages the unique properties of HPE and correlation indices to provide more supervision on active learning and pseudo-labelling during training.
>
> The integration of these components - HPE, correlation-aware pseudo-labeling, central correlation loss, and weighted cross-entropy loss - creates a cohesive framework specifically designed for SFADA.
>
> We further explain these distinctions and innovations in the revised manuscript as follows:
>
> “The weights $w(x_t)$ are determined based on whether the sample was part of the actively selected informative set or was assigned a pseudo-label. For actively selected samples, the weights incorporate their combined scores $U_i$. For pseudo-labeled samples, the weights are determined by the magnitude of their similarity score $z_{i,c}$ to their assigned class centroid. This weighting scheme ensures that highly confident pseudo-label assignments have more influence during training.” (Line 345-353)
>
> “B.4 Discussions
>
> The proposed ProULearn framework represents a significant advancement in SFADA through several key innovations. First, our HPE mechanism, combined with correlation-based feature relationships, introduces a novel approach to sample selection that considers both global data structure and local feature distributions, enabling more reliable identification of informative samples and pseudo-label assignment. Unlike traditional methods that rely heavily on initial distributions or require progressive labeling during training, ProULearn performs one-time sample selection before the adaptation process, making it particularly practical for real-world applications where continuous human annotation is infeasible. Second, our correlation index fundamentally differs from conventional distance metrics by capturing feature-level distribution relationships rather than spatial proximity, making it more robust to domain shifts. This correlation-based approach, when integrated with HPE, has demonstrated superior performance compared to traditional clustering methods. Furthermore, the newly proposed central correlation loss and weighted cross-entropy loss incorporate these components, creating more compact and discriminative class distributions. This cohesive integration of novel components forms a comprehensive framework tailored for SFADA challenges, while the underlying principles of our sample selection strategy offer broader applicability for various computer vision tasks where identifying informative samples or features is crucial.” (Line 815-831)
>
> **[to be continued in the next response window]**

---

> ### Author Response · Authors · 2024-11-18
>
> **[continuing from the previous response window]**
>
> **Comment**: The Correlation index needs clearer explanation. Does “k-th feature of sample x” mean the value of the k-th channel in the extracted feature of sample x? What is d in Eq.2? The notation k are repeatedly used through Eq.2 and 3.
>
> **Response**: Yes, after processing by the backbone network, an image is transformed into a 1D feature embedding. When we refer to "k-th feature of sample x," it means the k-th channel value in this extracted feature embedding, which is obtained after the average pooling operation of the k-th feature map.
>
> In Eq. 2, d represents the dimension of the feature embedding, which equals the number of channels in the network's feature output. As for the notation in Eq. 2 and 3, we use different cases to distinguish between different concepts: uppercase K denotes the number of nearest neighbors considered, while lowercase k serves as an index value in different contexts. For clarity, these notations are now listed in Table 5 in the Appendix. To avoid potential confusion, we have revised our notation in the manuscript of Eq. 2 and added the following explanation:
>
> “where $D$ is the dimension of sample $x_i$’s feature embedding $f(x_i)$, and $f(x_{i})^d$ is the $d$-th feature of sample $x_i$” (Line 257)
>
>
> **Comment**: The authors mention “computing the correlation matrix between all pairs of samples.” Could this process introduce high computation or storage burden during training?
>
> **Response**: Our correlation index calculation has similar computational requirements to L2 distance calculation, both requiring O(N²) for calculating the full similarity matrix. However, this computation is performed only once before training to select active samples, not during the training process itself. This one-time computation cost is insignificant compared to the iterative training process of deep neural networks. For example, for the largest VisDA dataset, our algorithm only takes about 1.4 minutes for the whole sample selection process with Intel i5-12500 CPU, RTX A5000 GPU, and 32 GB memory, which is manageable under modern devices.

---

> ### Author Response · Authors · 2024-11-26
>
> **Comment:** Thank you for your effort and response. While I don't fully agree with your statement that only using 7 domain shifts for the DomainNet-126 dataset is a common setting in SFDA and SFADA, I still appreciate the additional experiments provided. However, I am puzzled by an inconsistency in the experimental results: Table 3 and Table 8 show identical results despite using different labeled sample ratios (5% vs 10%). I noticed this is also the case in the revised PDF. This seems unusual, especially given your description in Appendix E.1 which states "On the larger-scale DomainNet-126 dataset with the same 10% budget (Table 8), our method maintains superior performance at 78.6% average accuracy" - suggesting this isn't a simple typographical error.
>
> Additionally, I noticed that the caption of Table 3 and the corresponding data (1.9%) in Section 4.2 have not been updated yet.
>
> **Response:** Thank you very much for your careful review which has helped improve the quality of our paper. We sincerely apologize for the inconsistency you identified between Tables 3 and 8. This error occurred when we copied the format from Table 3 and inadvertently failed to update the content with the new 10% labeling results. We have now corrected this error in Table 8 with the proper 10% labeling results as below and in the revised manuscript:
>
> Table 8
> | Categories | Method | SF | R→C | R→P | S→R | P→C | S→C | P→S | C→S | S→P | R→S | P→R | C→P | C→R | Avg |
> |-|-|:-:|-|-|-|-|-|-|-|-|-|-|-|-|-|
> | SFADA | MHPL† | ✓ | 82.2 | 79.5 | 90.3 | 82.2 | 83.7 | 77.1 | 76.2 | 78.5 | 75.1 | 90.9 | 77.4 | 90.2 | 81.9 |
> | | ProULearn | ✓ | 83.0 | 80.4 | 90.1 | 82.7 | 84.5 | 77.8 | 77.3 | 79.8 | 76.5 | 90.8 | 78.9 | 90.0 | **82.7** |
>
> We have also updated the caption of Table 3 and the corresponding data in Section 4.2 (which should be 1.6%) in the manuscript to reflect the correct performance difference. The core conclusions of our work remain unchanged. Thank you again for bringing this to our attention!

---

### Official Review · Reviewer_2XFN · 2024-11-04

**Soundness:** 3
**Presentation:** 3
**Contribution:** 2
**Rating:** 3
**Confidence:** 5

**Summary:**

In this paper, the authors present ProULearn, a Propensity-driven Uncertainty Learning framework for source-free active domain adaptation (SFADA). This approach tackles key challenges in SFADA with innovative techniques for sample selection and model adaptation, including a homogeneity propensity estimation mechanism and correlation relationship learning. ProULearn identifies informative samples while reducing the impact of noisy outliers and locates these samples before training begins, eliminating the need for human labels during adaptation. This approach, combined with a training strategy that results in more compact and discriminative class distributions in the target domain, enhances the practicality of domain adaptation.

**Strengths:**

1. The paper is well written and clear.

2. Extensive experiments done in order to validate the claim.

**Weaknesses:**

1. The main contribution of the paper, as I understand it, is the efficient selection of samples for labeling, similar to standard active learning approaches. I do not see any component of this method specifically tailored for SFADA; instead, it appears to be a generic sampling technique applicable to any active learning-based adaptation. In this regard, I find limited novelty in both the problem statement and the proposed method.

2. Given that sampling is a key strength of this method, more comparisons are necessary beyond DBSCAN or K-means. For instance, how would the results differ if sampling were based solely on entropy? I believe there are additional sampling techniques in the literature that could also be tested.

3. The experiments are insufficient. The authors should compare their approach with the most recent ECCV work for SFADA, [A], which seems to achieve better results than those reported in this paper.

[A] Learn from the Learnt: Source-Free Active Domain Adaptation via Contrastive Sampling and Visual Persistence

**Questions:**

1. In the case of the MHPL method (Active Source-Free Domain Adaptation), the reported number in the original paper is higher than what is reported here. For example, the Office-Home results in the MHPL paper are 79.1, while this paper reports 78.4 under the same experimental settings. What accounts for this discrepancy? I understand that results may sometimes differ when reproducing them from the original code, but if that is the case, it should be explicitly mentioned in the paper.

---

> ### Author Response · Authors · 2024-11-18
>
> **Comment**: The main contribution of the paper, as I understand it, is the efficient selection of samples for labeling, similar to standard active learning approaches. I do not see any component of this method specifically tailored for SFADA; instead, it appears to be a generic sampling technique applicable to any active learning-based adaptation. In this regard, I find limited novelty in both the problem statement and the proposed method.
>
> **Response**: Our contribution is multi-faceted and specifically designed for SFADA challenges while offering broader implications for the field.
>
> First, our Propensity-driven Uncertainty Learning (ProULearn) framework introduces a novel sample selection strategy that addresses a critical challenge in active learning - the need to identify informative samples. Different from traditional clustering methods that rely heavily on initial distributions, our HPE approach evaluates data structure at both local and global scales simultaneously by constructing an ensemble of separation trees. Each tree captures a different perspective of the data density patterns, making it particularly effective for informative sample selection.
>
> Second, our method offers a significant practical advantage over existing SFADA approaches. While some current methods require progressive labelling during training (as noted in lines 84-85), the proposed ProULearn innovatively performs sample selection before the adaptation process begins. This design directly addresses the real-world constraints of SFADA where continuous human annotation during training may be impractical or impossible.
>
> Third, we have developed a novel comprehensive SFADA framework that goes beyond just sample selection. The central correlation loss (as in Eq. 9) works in conjunction with our sample selection strategy to create more compact and discriminative class distributions during adaptation. This loss function, combined with our weighted cross-entropy loss where weights are derived from the proposed homogeneity scores and correlation indices, forms an integrated framework specifically tailored for SFADA challenges.
>
> While we agree that our proposed sample selection strategy could be beneficial for other active learning scenarios - which we view as a strength rather than a limitation - the complete ProULearn framework is specifically designed to address the unique challenges of SFADA.
>
> We further clarify our proposed ProULearn method's novelty and contribution in the revised manuscript as:
>
> “B.4 Discussions
>
> The proposed ProULearn framework represents a significant advancement in SFADA through several key innovations. First, our HPE mechanism, combined with correlation-based feature relationships, introduces a novel approach to sample selection that considers both global data structure and local feature distributions, enabling more reliable identification of informative samples and pseudo-label assignment. Unlike traditional methods that rely heavily on initial distributions or require progressive labeling during training, ProULearn performs one-time sample selection before the adaptation process, making it particularly practical for real-world applications where continuous human annotation is infeasible. Second, our correlation index fundamentally differs from conventional distance metrics by capturing feature-level distribution relationships rather than spatial proximity, making it more robust to domain shifts. This correlation-based approach, when integrated with HPE, has demonstrated superior performance compared to traditional clustering methods. Furthermore, the newly proposed central correlation loss and weighted cross-entropy loss incorporate these components, creating more compact and discriminative class distributions. This cohesive integration of novel components forms a comprehensive framework tailored for SFADA challenges, while the underlying principles of our sample selection strategy offer broader applicability for various computer vision tasks where identifying informative samples or features is crucial.” (Line 815-831)
>
> **[to be continued in the next response window]**

---

> ### Author Response · Authors · 2024-11-18
>
> **[continuing from the previous response window]**
>
> **Comment**: Given that sampling is a key strength of this method, more comparisons are necessary beyond HDBSCAN or K-means. For instance, how would the results differ if sampling were based solely on entropy? I believe there are additional sampling techniques in the literature that could also be tested.
>
> **Response**: Thank you for your comment. We would like to clarify that entropy serves as a measurement metric rather than a standalone sampling technique. To address this comment, we have expanded our evaluation to include both entropy-based measurement and PageRank centrality for guiding the sample selection process. These methods represent fundamentally different approaches to quantifying sample importance, with entropy focusing on prediction uncertainty while PageRank considers sample connectivity within the feature space. The following comparative results have been added to our manuscript (Table 4).
>
> | Method | A→D | A→W | D→A | D→W | W→A | W→D | Avg |
> |--------|-----|-----|-----|-----|-----|-----|-----|
> | ProULearn+*HDBSCAN* | 96.8 | 97.2 | 83.5 | 98.4 | 81.4 | 99.6 | 92.8 |
> | ProULearn+*K-means* | 97.0 | 96.9 | 81.0 | 98.9 | 81.5 | 99.6 | 92.4 |
> | ProULearn+*Entropy* | 95.4 | 95.9 | 79.2 | 97.1 | 80.4 | 100.0 | 91.3 |
> | ProULearn+*PageRank* | 94.6 | 94.5 | 80.3 | 97.6 | 81.4 | 100.0 | 91.4 |
> | **ProULearn+*HPE*** | 99.2 | 96.3 | 84.4 | 99.4 | 84.4 | 100.0 | **94.0** |
>
>
> **Comment**: The experiments are insufficient. The authors should compare their approach with the most recent ECCV work for SFADA, [A], which seems to achieve better results than those reported in this paper.
> [A] Learn from the Learnt: Source-Free Active Domain Adaptation via Contrastive Sampling and Visual Persistence
>
> **Response**: Thank you for bringing attention to the recent paper from ECCV. While the paper reports impressive results using 10% active labels on Office31 and Office-Home datasets, our method focuses on and achieves better performance with 5% active labels.
>
> Though their source code remains unreleased, preventing direct verification in our training environment, we have conducted additional experiments using 10% active labels to enable thorough comparison. The performance data from these experiments demonstrates our method's effectiveness and superiority across both 5% and 10% labeling scenarios. We have also evaluated our approach on the more challenging DomainNet-126 dataset, which was not explored in their work. The comprehensive comparison tables containing these new results are added in the revised manuscript (see Tables 1 and 2).
>
>
> **Comment**: In the case of the MHPL method (Active Source-Free Domain Adaptation), the reported number in the original paper is higher than what is reported here. For example, the Office-Home results in the MHPL paper are 79.1, while this paper reports 78.4 under the same experimental settings. What accounts for this discrepancy? I understand that results may sometimes differ when reproducing them from the original code, but if that is the case, it should be explicitly mentioned in the paper.
>
> **Response**: We acknowledge the discrepancy between our reported MHPL results and those in their original paper. In our study, we reproduce the MHPL results under the same experimental environment as our method to ensure fair comparison. As indicated by the symbol (†) in our paper (line 408), these results are obtained using the original code from published papers.
> To provide complete transparency, we observe that MHPL has different versions available online with varying reported results. Our reproduced results may differ from the published numbers due to hardware environment variations and Python Library versions.
>
> To support reproducibility and facilitate future comparisons, we denote all reproduced results with the symbol (†) in our tables. We clarify this in the revised manuscript by adding:
>
> "The best average accuracy is marked in bold. '†' indicates that results are obtained using the original code from the published papers. The discrepancy between the reproduced results and the results reported in the published paper may be due to hardware environment differences." (Line 408-410)

---

### Note · Authors · 2025-01-22

I have read and agree with the venue's withdrawal policy on behalf of myself and my co-authors.